# NRGBoost: Energy-Based Generative Boosted Trees

## Abstract

Despite the rise to dominance of deep learning in unstructured data domains, tree-based methods such as Random Forests (RF) and Gradient Boosted Decision Trees (GBDT) are still the workhorses for handling discriminative tasks on tabular data. We explore generative extensions of these popular algorithms with a focus on explicitly modeling the data density (up to a normalization constant), thus enabling other applications besides sampling. As our main contribution we propose an effective energy-based generative boosting algorithm that is analogous to the second order boosting algorithm implemented in popular packages like XGBoost. We show that, despite producing a generative model capable of handling inference tasks over any input variable, our proposed algorithm can achieve similar discriminative performance to GBDT algorithms on a number of real world tabular datasets and outperform competing approaches for sampling.

## 1 Introduction

Generative models have achieved tremendous success in computer vision and natural language processing, where the ability to generate synthetic data guided by user prompts opens up many exciting possibilities. While generating synthetic table records does not necessarily enjoy the same wide appeal, this problem has still received considerable attention as a potential avenue for bypassing privacy concerns when sharing data. Estimating the data density, $p(\mathbf{x})$, is another typical application of generative models which enables a host of different use cases that can be particularly interesting for tabular data. Unlike discriminative models which are trained to perform inference over a single target variable, density models can be used more flexibly for inference over different variables or for out of distribution detection. They can also handle inference with missing data in a principled way by marginalizing over unobserved variables.

The development of generative models for tabular data has mirrored its progression in computer vision with many of its Deep Learning (DL) approaches being adapted to the tabular domain [Jordon et al., 2018, Xu et al., 2019, Engelmann and Lessmann, 2020, Fan et al., 2020, Zhao et al., 2021, Kotelnikov et al., 2022]. Unfortunately, these methods are only useful for sampling as they either don't model the density explicitly or can't evaluate it due to untractable marginalization over high dimensional latent variable spaces. Furthermore, despite growing in popularity, DL has still failed to displace tree-based ensemble methods as the tool of choice for handling tabular discriminative tasks with gradient boosting still being found to outperform neural-network-based methods in many real world datasets [Grinsztajn et al., 2022, Borisov et al., 2022a].

While there have been recent efforts to extend the success of tree-based models to generative modeling [Correia et al., 2020, Wen and Hang, 2022, Nock and Guillame-Bert, 2022, Watson et al., 2023, Nock and Guillame-Bert, 2023, Jolicoeur-Martineau et al., 2023], we find that direct extensions of Random Forests (RF) and Gradient Boosted Decision Tree (GBDT) are still missing. It is this gap that we try to address, seeking to keep the general algorithmic structure of these popular algorithms

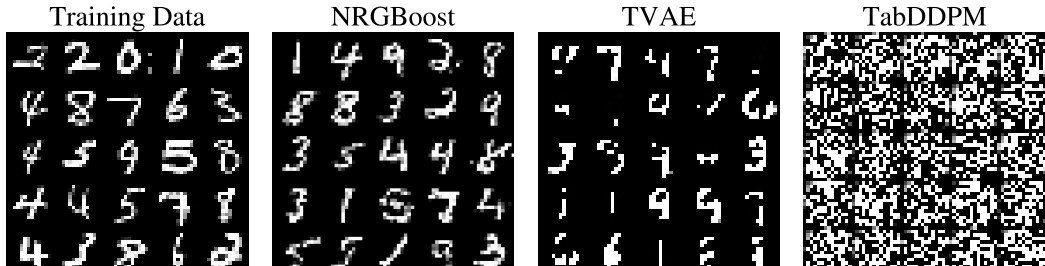

Figure 1: Downsampled MNIST samples generated by NRGBoost and two tabular DL methods.

but replacing the optimization of their discriminative objective with a generative counterpart. Our main contributions in this regard are:

- Proposing NRGBoost, a novel energy-based generative boosting model that, analogously to the boosting algorithms implemented in popular GBDT packages, is trained to maximize a local second order approximation to the likelihood at each boosting round.
- Proposing an approximate sampling algorithm to speed up the training of any tree-based multiplicative generative boosting model.
- Exploring the use of bagged ensembles of Density Estimation Trees (DET) [Ram and Gray, 2011] with feature subsampling as the generative counterpart to RF.

The longstanding popularity of GBDT models in machine learning practice can, in part, be attributed to the strength of its empirical results and the efficiency of its existing implementations. We therefore focus on an experimental evaluation in real world datasets spanning a range of use cases, number of samples and features. We find that, on smaller datasets, our implementation of NRGBoost can be trained in a few minutes on a mid-range consumer CPU and achieve similar discriminative performance to a standard GBDT model while also being able to generate samples that are generally harder to distinguish from real data than state of the art neural-network-based models.

## 2  Energy Based Models

An Energy-Based Model (EBM) parametrizes the logarithm of a probability density function directly (up to an unspecified normalizing constant):

$$q_f(\mathbf{x}) = \frac{\exp\left(f(\mathbf{x})\right)}{Z[f]}. \tag{1}$$

Here $f(\mathbf{x}) : \mathcal{X} \to \mathbb{R}$ is a real function over the input domain.[1] We will avoid introducing any parametrization, instead treating the function $f \in \mathcal{F}(\mathcal{X})$ lying in an appropriate function space over the input space as our model parameter directly. $Z[f] = \sum_{\mathbf{x} \in \mathcal{X}} \exp\left(f(\mathbf{x})\right)$, known as the partition function, is then a functional of $f$ giving us the necessary normalizing constant.

This is the most flexible way one could represent a probability density function making essentially no compromises on its structure. The downside to this is that for most interesting choices of $\mathcal{F}$, computing or estimating this normalizing constant is untractable which makes training these models difficult. Their unnormalized nature however does not prevent EBMs from being useful in a number of applications besides sampling. Performing inference over a small enough subset of variables requires only normalizing over the set of their possible values and for anomaly or out of distribution detection, knowledge of the normalizing constant is not necessary.

One common way to train an energy-based model to approximate a data generating distribution, $p(\mathbf{x})$, is to minimize the Kullback-Leibler divergence between $p$ and $q_f$, or equivalently, maximize the expected log likelihood functional:

$$L[f] = \mathbb{E}_{\mathbf{x} \sim p} \log q_f(\mathbf{x}) = \mathbb{E}_{\mathbf{x} \sim p} f(\mathbf{x}) - \log Z[f] \tag{2}$$

---

[1]We will assume that $\mathcal{X}$ is finite and discrete to simplify the notation and exposition but everything is applicable to bounded continuous input spaces, replacing the sums with integrals as appropriate.

This optimization is typically carried out by gradient descent over the parameters of $f$, but due to the untractability of the partition function, one must rely on Markov Chain Monte Carlo (MCMC) sampling to estimate the gradients [Song and Kingma, 2021].

## 3 NRGBoost

Expanding the increase in log-likelihood in equation 2 due to a variation $\delta f$ around an energy function $f$ up to second order we have

$$L[f + \delta f] - L[f] \approx \mathbb{E}_{\mathbf{x} \sim p} \delta f(\mathbf{x}) - \mathbb{E}_{\mathbf{x} \sim q_f} \delta f(\mathbf{x}) - \frac{1}{2} \text{Var}_{\mathbf{x} \sim q_f} \delta f(\mathbf{x}) =: \Delta L_f[\delta f] \,. \tag{3}$$

The $\delta f$ that maximizes this quadratic approximation should thus have a large positive difference between the expected value under the data and under $q_f$ while having low variance under $q_f$. We note that just like the original log-likelihood, this Taylor expansion is invariant to adding an overall constant to $\delta f$. This means that, in maximizing equation 3 we can consider only functions that have zero expectation under $q_f$ in which case we can simplify $\Delta L_f[\delta f]$ as

$$\Delta L_f[\delta f] = \mathbb{E}_{\mathbf{x} \sim p} \delta f(\mathbf{x}) - \frac{1}{2} \mathbb{E}_{\mathbf{x} \sim q_f} \delta f^2(\mathbf{x}) \,. \tag{4}$$

We thus formulate our boosting algorithm as modelling the data density with an additive energy function. At each boosting iteration we improve upon the current energy function $f_t$ by finding an optimal step $\delta f_t^*$ that maximizes $\Delta L_{f_t}[\delta f]$

$$\delta f_t^* = \arg \max_{\delta f \in \mathcal{H}_t} \Delta L_{f_t}[\delta f] \,, \tag{5}$$

where $\mathcal{H}_t$ is an appropriate space of functions (satisfying $\mathbb{E}_{\mathbf{x} \sim q_{f_t}} \delta f(\mathbf{x}) = 0$ if equation 4 is used). The solution to this problem can be interpreted as a Newton step in the space of energy functions. Because for an energy-based model, the Fisher Information matrix with respect to the energy function and the hessian of the expected log-likelihood are the same, we can also interpret the solution to equation 5 as a natural gradient step (see the Appendix A). This approach is essentially analogous to the second order step implemented in modern discriminative gradient boosting libraries such as XGBoost [Chen and Guestrin, 2016] and LightGBM [Ke et al., 2017] and which can be traced back to Friedman et al. [2000].

In updating the current iterate, $f_{t+1} = f_t + \alpha_t \cdot \delta f_t^*$, we scale $\delta f_t^*$ by an additional scalar step-size $\alpha_t$. This can be interpreted as a globalization strategy to account for the fact that the quadratic approximation in equation 3 is not necessarily valid over large steps in function space. A common strategy in nonlinear optimization would be to select $\alpha_t$ via a line search based on the original log-likelihood. Common practice in discriminative boosting however is to interpret this step size as a regularization parameter and to select a fixed value in $]0, 1]$ with (more) smaller steps typically outperforming fewer larger ones when it comes to generalization. We choose to adopt a hybrid strategy, first selecting an optimal step size by line search and then shrinking it by a fixed factor. We find that this typically accelerates convergence allowing the algorithm to take comparatively larger steps that increase the likelihood in the initial phase of boosting. For a starting point, $f_0$, we can choose the logarithm of any probability distribution over $\mathcal{X}$ as long as it is easy to evaluate. Sensible choices are a uniform distribution (i.e., $f \equiv 0$), the product of marginals for the training set, or any mixture distribution between these two.

### 3.1 Weak Learners

As a weak learner we will consider functions defined by trees over the input space. I.e., letting $\bigcup_{j=1}^J X_j = \mathcal{X}$ be the partitioning of the input space induced by the leaves of a binary tree whose internal nodes represent a split along one dimension into two disjoint partitions, we take as $\mathcal{H}$ the set of functions such as

$$\delta f(\mathbf{x}) = \sum_{j=1}^J w_j \mathbf{1}_{X_j}(\mathbf{x}) \,, \tag{6}$$

where $\mathbf{1}_X$ denotes the indicator function of a subset $X$ and $w_j$ are values associated with each leaf $j \in [1..J]$. In a standard decision tree these values would typically encode an estimate of

$p(y|\mathbf{x} \in X_j)$, with $y$ being a special *target* variable that is never considered for splitting. In our generative approach they encode unconditional densities (or more precisely energies) over each leaf's support and every variable can be used for splitting. Note that our functions $\delta f$ are thus parametrized by the values $w_j$ as well the structure of the tree and the variables and values for the split at each node which ultimately determine the $X_j$. We omit these dependencies for brevity.

Replacing the definition in equation 6 in our objective (equation 4) we get the following optimization problem to find the optimal decision tree:

$$\max_{w_1,\ldots,w_J,X_1,\ldots,X_J} \quad \sum_{j=1}^{J} \left( w_j P(X_j) - \frac{1}{2} w_j^2 Q_f(X_j) \right)$$
$$\text{s.t.} \quad \sum_{j=1}^{J} w_j Q_f(X_j) = 0 \,, \tag{7}$$

where $P(X_j)$ and $Q_f(X_j)$ denote the probability of the event $\mathbf{x} \in X_j$ under the respective distribution and the constraint ensures that $\delta f$ has zero expectation under $q_f$. With respect to the leaf weights this is a quadratic program whose optimal solution and objective values are respectively given by

$$w_j^* = \frac{P(X_j)}{Q_f(X_j)} - 1 \,, \qquad \Delta L_f^*(X_1,\ldots,X_J) = \frac{1}{2} \left( \sum_{j=1}^{J} \frac{P^2(X_j)}{Q_f(X_j)} - 1 \right) \,. \tag{8}$$

Because carrying out the maximization of this optimal value over the tree structure that determines the $X_j$ is hard, we approximate its solution by greedily growing a tree that maximizes it when considering how to split each node individually. A parent leaf with support $X_P$ is thus split into 2 child leaves, with disjoint support, $X_L \cup X_R = X_P$, so as to maximize over all possible partitionings along a single dimension, $\mathcal{P}(X_P)$, the following objective:

$$\max_{X_L,X_R \in \mathcal{P}(X_P)} \frac{P^2(X_L)}{Q_f(X_L)} + \frac{P^2(X_R)}{Q_f(X_R)} - \frac{P^2(X_P)}{Q_f(X_P)} \,. \tag{9}$$

Note that when using parametric weak learners, computing a second order step would typically involve solving a linear system with a full Hessian. As we can see, this is not the case when the weak learners are decision trees where the optimal value to assign to a leaf $j$ does not depend on any information from other leaves and, likewise, the optimal objective value is a sum of terms, each depending only on information from a single leaf. This would have not been the case had we tried to optimize the likelihood functional in Equation 2 directly instead of its quadratic approximation.

## 3.2 Sampling

To compute the leaf values in equation 8 and the splitting criterion in equation 9 we would have to know $P(X)$ and be able to compute $Q_f(X)$ which is infeasible due to the untractable normalization constant. We therefore estimate these quantities, with recourse to empirical data for $P(X)$, and to samples approximately drawn from the model with MCMC. Because even if the input space is not partially discrete, $f$ is still discontinuous and constant almost everywhere we can't use gradient based samplers and therefore rely on Gibbs sampling instead. This only requires evaluating each $f_t$ along one dimension at a time, while keeping all others fixed which can be computed efficiently for a tree by traversing it only once. However, since at boosting iteration $t$ our energy function is a sum of $t$ trees, this computation scales linearly with the iteration number. This makes the overall time spent sampling quadratic in the number of iterations and thus precludes us from training models with a large number of trees.

In order to reduce the burden associated with this sampling, which can dominate the runtime of training the model, we propose a new sampling approach that leverages the cumulative nature of boosting. The intuition behind this approach is that the set of samples used in the previous boosting round are (approximately) drawn from a distribution that is already close to the new model distribution. It could therefore be helpful to keep some of those samples, especially those that conform the best to the new model. Rejection sampling allows us to do just that. The boosting update in terms of the densities takes the following multiplicative form:

$$q_t(\mathbf{x}) = k_t \, q_{t-1}(\mathbf{x}) \exp\left( \alpha_t \delta f_t(\mathbf{x}) \right) \,. \tag{10}$$

Here, $k$ is an unknown multiplicative constant and since $\delta f_t$ is given by a tree, we can easily bound the exponential factor by finding the leaf with the largest value. We can therefore use the previous model, $q_{t-1}(\mathbf{x})$, as a proposal distribution for which we already have a set of samples and keep each sample, $\mathbf{x}$, with an acceptance probability of:

$$p_{accept}(\mathbf{x}) = \exp\left[\alpha_t\left(\delta f_t(\mathbf{x}) - \max_{\mathbf{x}} \delta f_t(\mathbf{x})\right)\right].\qquad(11)$$

We note that knowledge of the constant $k_t$ is not necessary to compute this acceptance probability. After removing samples from the pool, we can use Gibbs sampling to draw a new set of samples in order to keep a fixed total number of samples per round of boosting. Note also that $q_0$ is typically a simple model for which we can both directly evaluate the desired quantities (i.e., $Q_0(X)$ for a given partition $X$) and cheaply draw exact samples from. As such, no sampling is required for the first iteration of boosting and for the second we can draw exact samples from $q_1$ with rejection sampling using $q_0$ as a proposal distribution.

This approach works better when either the range of $f_t$ is small or when the step sizes $\alpha_t$ are small as this leads to larger acceptance probabilities. Note that in practice it can be helpful to independently refresh a fixed fraction samples, $p_{refresh}$, at each round of boosting in order to encourage more diverse samples between rounds. This can be accomplished by keeping each sample with a probability $p_{accept}(\mathbf{x})(1 - p_{refresh})$ instead.

### 3.3 Regularization

The simplest way to regularize a boosting model is to stop training when overfitting is detected by monitoring a suitable performance metric on a validation set. For NRGBoost this could be the increase in log-likelihood at each boosting round. However, estimating this quantity would require drawing additional validation samples from the model (see Appendix A). An alternative viable validation strategy which needs no additional samples is to simply monitor a discriminative performance metric (over one or more variables). This essentially amounts to monitoring the quality of $q_f(x_i|\mathbf{x}_{-i})$ instead of the full $q_f(\mathbf{x})$.

Besides early stopping, the decision trees themselves can be regularized by limiting the depth or total number of leaves of each tree. Additionally we can rely on other strategies such as disregarding splits that would result in a leaf with too little training data, $P(X)$, model data, $Q_f(X)$, volume $V(X)$ or too high of a ratio between training and model data $P(X)/Q_f(X)$. We found the latter to be the most effective of these, not only yielding better generalization performance than other approaches, but also having the added benefit of allowing us to lower bound the acceptance probability of our rejection sampling scheme.

## 4 Density Estimation Trees and Density Estimation Forests

Density Estimation Trees (DET) were proposed by Ram and Gray [2011] as an alternative to histograms and kernel density estimation but have received little attention as generative models for sampling or other applications. They model the density function as a constant value over the support of each leaf in a binary tree, $q = \sum_{j=1}^{J} \frac{\hat{P}(X_j)}{V(X_j)}\mathbf{1}_{X_j}$, with $\hat{P}(X)$ being an empirical estimate of probability of the event $\mathbf{x} \in X$ and $V(X)$ denoting the volume of $X$. Note that it is possible to draw an exact sample from this type of model by randomly selecting a leaf, $j \in [1..J]$, given probabilities $\hat{P}(X_j)$, and then drawing a sample from a uniform distribution over $X_j$.

To fit a DET, Ram and Gray [2011] propose optimizing the Integrated Squared Error (ISE) between the data and model distributions which, following a similar approach to Section 3.1, leads the following optimization problem when considering how to split a leaf node:

$$\max_{X_L, X_R \in \mathcal{P}(X_P)} D(P(X_L), V(X_L)) + D(P(X_R), V(X_R)) - D(P(X_P), V(X_P)).\qquad(12)$$

For the ISE, $D$ should be taken as the function $D_{ISE}(P, V) = P^2/V$ which leads to a similar splitting criterion to Equation 12 but replacing the previous model's distribution with the volume measure $V$ which can be interpreted as the uniform distribution on $\mathcal{X}$ (up to a multiplicative constant).

**Maximum Likelihood** Often generative models are trained to maximize the likelihood of the observed data. This was left for future work in Ram and Gray [2011] but, as we show in Appendix B, can be accomplished by replacing the $D$ in Equation 12 with $D_{KL}(P, V) = P \log (P/V)$. This choice of minimization criterion can be seen as analogous to the choice between Gini impurity and Shannon entropy in the computation of the information gain in decision trees.

**Bagging and Feature Subsampling** Following the common approach in decision trees, Ram and Gray [2011] suggest the use of pruning for regularization of DET models. Practice has however evolved to prefer bagging as a form of regularization rather than relying on single decision trees. We employ same principle to DETs by fitting many trees on bootstrap samples of the data. We also adopt the common practice from Random Forests of randomly sampling a subset of features to consider when splitting any leaf node in order to encourage independence between the different trees in the ensemble. The ensemble model, which we call *Density Estimation Forests* (DEF) in the sequence, is thus an additive mixture of DETs with uniform weights, therefore still allowing for normalized density computation and exact sampling.

## 5 Related Work

**Generative Boosting** Most prior work on generative boosting focuses on unstructured data and the use of parametric weak learners and is split between two approaches: (i) Additive methods that model the density function as an additive mixture of weak learners such as Rosset and Segal [2002], Tolstikhin et al. [2017]. (ii) Those that take a multiplicative approach modeling the density function as an unnormalized product of weak learners. The latter is equivalent to the energy based approach that writes the energy function (log density) as an additive sum of weak learners. Welling et al. [2002] in particular also approach boosting from the point of view of functional optimization of the likelihood or the logistic loss of an energy-based model. However, they rely on a first order local approximation of the objective since they focus on parametric weak learners such as restricted boltzman machines for which a second order step would be impractical.

**Greedy Multiplicative Boosting** Another more direct multiplicative boosting framework was first proposed by Tu [2007]. At each boosting round a discriminative classifier is trained to distinguish between empirical data and data generated by the current model by estimating the likelihood ratio $p(\mathbf{x})/q_t(\mathbf{x})$. This estimated ratio is used as a direct multiplicative factor to update the current model $q_t$ (after being raised to an appropriate step size). In ideal conditions this greedy procedure would converge in a single iteration if a step size of 1 would be used. While Tu [2007] does not prescribe a particular choice of classifier to use, Grover and Ermon [2017] proposes a similar concept where the ratio is estimated based on an adversarial bound for an $f$-divergence and Cranko and Nock [2019] provides additional analysis on this method. In Appendix C we dive deeper into the differences between NRGBoost and this approach when it is adapted to use trees as weak learners. We note, however, that the main difference is that NRGBoost attempts to update the current density proportionally to an exponential of the ratio, $\exp(\alpha_t \cdot p(x)/q_t(x))$, instead of the ratio directly.

**Tree-Based Density Modelling** Other authors have proposed tree-based density models similar to DET [Nock and Guillame-Bert, 2022] or additive mixtures of tree-based models [Correia et al., 2020, Wen and Hang, 2022, Watson et al., 2023] but perhaps surprisingly, the natural idea of creating an ensemble of DET models through bagging has not been explored before as far as we are aware. Two distinguishing features of some of these alternative approaches are: (i) Unlike DETs, the partitioning of each tree is not driven directly by a density estimation goal. Correia et al. [2020] leverages a standard discriminative Random Forest, therefore giving special treatment to a particular input variable whose conditional estimation drives the choice of partitions and Wen and Hang [2022] proposes using a mid-point random tree partitioning. (ii) Besides modelling the density function as uniform at the leaf of each tree, other authors propose leveraging more complex models [Correia et al., 2020, Watson et al., 2023] which can allow for the use of trees that are more representative with a smaller number of leaves. (iii) Nock and Guillame-Bert [2022] and Watson et al. [2023] both propose generative adversarial frameworks where the generator and discriminator are both a tree or an ensemble of trees respectively. Note that, unlike with boosting, in these approaches the new model doesn't add to the previous one but replaces it instead.

Table 1: Single variable inference results. The reported values are the averages over 5 cross-validation folds. The corresponding sample standard deviations are reported in Appendix G.

| | $R^2 \uparrow$ | | | AUC $\uparrow$ | | Accuracy $\uparrow$ | |
| | AB | CH | PR | AD | MBNE | MNIST | CT |
|---|---|---|---|---|---|---|---|
| XGBoost | 0.552 | 0.849 | 0.678 | 0.927 | 0.987 | 0.976 | 0.972 |
| RFDE | 0.071 | 0.340 | 0.059 | 0.862 | 0.668 | 0.302 | 0.681 |
| DEF (ISE) | 0.467 | 0.737 | 0.566 | 0.854 | 0.653 | 0.206 | 0.790 |
| DEF (KL) | 0.482 | 0.801 | 0.639 | 0.892 | 0.939 | 0.487 | 0.852 |
| NRGBoost | **0.547** | **0.850** | **0.676** | **0.920** | **0.974** | **0.966** | **0.949** |

**Other Recent Tree-Based approaches**  Nock and Guillame-Bert [2023] proposes a different ensemble approach where each tree does not have their own leaf values that get added or multiplied to produce the final density, but instead serve to collectively define the partitioning of the input space. To train such models the authors propose a boosting framework where, rather than adding a new tree to the ensemble at every iteration, the model is initialized with a fixed number of tree root nodes and each iteration adds a split to an existing leaf node. Finally Jolicoeur-Martineau et al. [2023] propose a diffusion model where a tree-based model (e.g., GBDT) is used to regress the score function. Being a diffusion model, however, means that computing densities is untractable.

## 6 Experiments

For our experiments we use 5 tabular datasets from the UCI Machine Learning Repository [Dheeru and Karra Taniskidou, 2017]: Abalone (AB), Physicochemical Properties of Protein Tertiary Structure (PR), Adult (AD), MiniBooNE (MBNE) and Covertype (CT) as well as the California Housing (CH) available through the Scikit-Learn package [Pedregosa et al., 2011]. We also include a downsampled version of MNIST (by 2x along each dimension) which allows us to visually assess the quality of individual samples, something that is generally not possible with structured tabular data, and provides an example of the performance that can be achieved in an unstructured dataset with many features that are correlated among themselves. More details about these datasets are given in Appendix E.

We split our experiments into two sections, the first to evaluate the quality of density models directly on a single variable inference task and the second to investigate the performance of our proposed models when used for sampling.

### 6.1  Single Variable Inference

In this section we test the ability of a generative model, trained to learn the density over all input variables, $q(\mathbf{x})$, to infer the value of a single one. I.e., we wish to test how good is its estimate of $q(x_i|\mathbf{x}_{-i})$. For this purpose we pick $x_i = y$ as the original target of the dataset, noting that the models that we train do not treat this variable in any special way, except for the selection of the best model in validation. As such, we would expect that the model's performance in inference over this particular variable is indicative of its strength on any other single variable inference task and also indicative of the quality of the full $q(\mathbf{x})$ from which the conditional probability estimate is derived.

We use XGBoost [Chen and Guestrin, 2016] as a baseline for what should be achievable by a very strong discriminative model. Note that this model is trained to maximize the discriminative likelihood, $\mathbb{E}_{\mathbf{x} \sim p} \log q(x_i|\mathbf{x}_{-i})$, directly, not wasting model capacity in learning other aspects of the full data distribution. As another generative baseline we use our own implementation of RFDE [Wen and Hang, 2022] which allows us to gauge the impact of the guided partitioning used in the DEF models over a random partitioning of the input space.

We use random search to tune the hyperparameters of the XGBoost model and a grid search to tune the most important hyperparameters of the generative density models. We employ 5-fold cross-validation, repeating the hyperparameter tuning on each fold for all datasets except for the largest one (CT) for which we report results on a single fold. For the full details of the experimental protocol please refer to Appendix F.

Table 2: ML Efficiency results. The reported values are the averages over 5 different datasets generated by the same model. The best methods for each dataset are in **bold** and methods whose difference is $< 2\sigma$ away from zero are underlined. The performance of XGBoost trained on the real data is also reported for reference.

| | $R^2 \uparrow$ | | | AUC $\uparrow$ | | Accuracy $\uparrow$ | |
| --- | --- | --- | --- | --- | --- | --- | --- |
| | AB | CH | PR | AD | MBNE | MNIST | CT |
| XGBoost | 0.554 | 0.838 | 0.682 | 0.927 | 0.987 | 0.976 | 0.972 |
| TVAE | 0.483 | 0.758 | 0.365 | 0.898 | 0.975 | 0.688 | 0.724 |
| TabDDPM | **0.539** | **0.807** | **0.596** | 0.910 | **0.984** | 0.579 | 0.818 |
| DEF (KL) | 0.450 | 0.762 | 0.498 | 0.892 | 0.943 | 0.230 | 0.753 |
| NRGBoost | 0.528 | 0.801 | 0.573 | **0.914** | 0.977 | **0.959** | **0.895** |

289  We find that NRGBoost performs better than the additive ensemble models (see Table 1) despite
290  producing more compact ensembles. It often achieves comparable performance to XGBoost on the
291  smaller datasets and with a small gap on the three larger ones. We note also that for the regression
292  datasets the generative models provide an estimate of the full conditional distribution over the target
293  variable rather than a point estimate like XGBoost. While there are other variants of discriminative
294  boosting that also provide an estimate of the aleatoric uncertainty [Duan et al., 2020], they rely on a
295  parametric assumption about $p(y|\mathbf{x})$ that needs to hold for any $\mathbf{x}$.

## 6.2   Sampling

297  In this section, we compare the sampling performance of our proposed methods to neural-network-
298  based methods TVAE [Xu et al., 2019] and TabDDPM [Kotelnikov et al., 2022] on two metrics.

**Machine Learning Efficiency**   The Machine Learning (ML) efficiency has been a popular way
300  to measure the quality of generative models for sampling [Xu et al., 2019, Kotelnikov et al., 2022,
301  Borisov et al., 2022b]. It relies on using samples from the model to train a discriminative model which
302  is then evaluated on the real data. Note that this is similar to the single variable inference performance
303  from Section 6.1. In fact, if the density model's support covers that of the full data, one would expect
304  the discriminative model to recover the generator's $q(y|\mathbf{x})$, and therefore its performance, in the limit
305  where infinite generated data is used to train it.

306  We use an XGBoost model (with the hyperparameters tuned in real data) as the discriminative model
307  and train it using a similar number of training and validation samples as in the original data. For
308  the density models, we generate samples from the best model found in the previous section and
309  for non-density models we select their hyperparameters by evaluating the ML Efficiency in the
310  real validation set. Note that this leaves the sampling models at a potential advantage since the
311  hyperparameter selection is based on the metric that is being evaluated rather than the direct inference
312  performance of the previous section.

**Discriminator Measure**   Similar to Borisov et al. [2022b] we test the capacity of a discriminative
314  model to distinguish between real and generated data. We use the original validation set as the real
315  part of the training data in order to avoid benefiting generative methods that overfit their original
316  training set. A new validation set is carved out of the original test set (20%) and used to tune the
317  hyperparameters of an XGBoost model which we use as our choice of discriminator, evaluating its
318  AUC on the remainder of the real test data.

319  We repeat all experiments 5 times, with 5 different generated datatsets from each model. Results are
320  reported in Tables 2 and 3 showing that (i) NRGBoost outperforms all other methods by substantial
321  margins in the discriminator measure except for the PR and the MBNE datasets. (ii) On the ML
322  Efficiency metric, TabDDPM outperforms NRGBoost by small margins on the small datasets which
323  could in part be explained by the denser hyperparameter tuning favouring models that perform
324  particularly well at inferring the target variable at the expense of the others. Nevertheless, NRGBoost
325  still significantly outperforms all other models on MNIST and CT. Its samples also look visually
326  similar to the real data in both the MNIST and California datasets (see Figures 1 and 2).

Table 3: Discriminator measure results. All results are the AUC of an XGBoost model trained to distinguish real from generated data an therefore lower means better. The reported values are the averages over 5 different datasets generated by the same model.

|          | AB       | CH       | PR       | AD       | MBNE     | MNIST    | CT       |
|----------|----------|----------|----------|----------|----------|----------|----------|
| TVAE     | 0.971    | 0.834    | 0.940    | 0.898    | 1.000    | 1.000    | 0.999    |
| TabDDPM  | 0.818    | 0.667    | **0.628**| 0.604    | **0.789**| 1.000    | 0.915    |
| DEF (KL) | 0.823    | 0.751    | 0.877    | 0.956    | 1.000    | 1.000    | 0.999    |
| NRGBoost | **0.625**| **0.574**| 0.631| **0.559**| 0.993 | **0.943**| **0.724**|

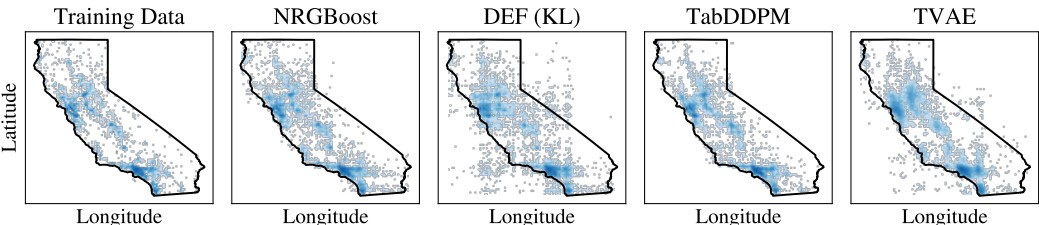

Figure 2: Joint histogram for the latitude and longitude for the California Housing dataset.

## 7 Discussion

While the additive tree models like DEF require no sampling to train and are easy to sample from, we find that in practice they require very deep trees to model the data well which, in turn, also requires using a large number of trees in the ensemble to regularize. In our experiments we found that their performance was often capped by the maximum number of leaves we allowed them to grow to ($2^{14}$).

In contrast, we find that NRGBoost is able to model the data better while using shallower trees and in fewer number. Its main downside is that it can only be sampled from approximately using more expensive MCMC and also requires sampling during the training process. While our fast Gibbs sampling implementation coupled with our proposed sampling approach were able to mitigate the slow training, making these models much more usable in practice they are still cumbersome to use for sampling due to autocorrelation between samples from the same Markov Chain. We argue however that unlike in image or text generation where fast sampling is necessary for an interactive user experience, this can be less of a concern for the task of generating synthetic datasets where the one time cost of sampling is not as important as faithfully capturing the data generating distribution.

We also find that tuning the hyperparameters of tree-based models is easier and less crucial than DL models for which many trials fail to produce a reasonable model. In particular we found NRGBoost to be rather robust, with different hyperparameters leading to small differences in performance.

Finally, we note that like any other machine learning models, generative models are susceptible to overfitting and are thus liable to leak information about their training data when generating synthetic samples. In this respect, we believe that NRGBoost offers better tools to monitor and control overfitting than other alternatives (see Section 3.3) but, still, due consideration for this risk must be taken into account when sharing synthetic data.

## 8 Conclusion

In this work, we extend the two most popular tree-based discriminative methods for use in generative modeling. We find that our boosting approach, in particular, offers generally good discriminative performance and better overall sampling performance than alternatives. We hope that these results encourage further research into generative boosting approaches for tabular data, in particular exploring other applications besides sampling that are enabled by density models.

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

## A  Additional Derivations

The expected log-likelihood for an energy-based model (EBM),

$$q_f(\mathbf{x}) = \frac{\exp\left(f(\mathbf{x})\right)}{Z[f]}, \tag{13}$$

is given by

$$L[f] = \mathbb{E}_{\mathbf{x}\sim p} \log q_f(\mathbf{x}) = \mathbb{E}_{\mathbf{x}\sim p} f(\mathbf{x}) - \log Z[f]. \tag{14}$$

The *first variation* of $L$ can be computed as

$$\delta L[f;g] := \left.\frac{dL[f + \epsilon g]}{d\epsilon}\right|_{\epsilon=0} = \mathbb{E}_{\mathbf{x}\sim p}\, g(\mathbf{x}) - \delta \log Z[f;g] = \mathbb{E}_{\mathbf{x}\sim p}\, g(\mathbf{x}) - \mathbb{E}_{\mathbf{x}\sim q_f}\, g(\mathbf{x}). \tag{15}$$

This is a linear functional of its second argument, $g$, and can be regarded as a directional derivative of $L$ at $f$ along a variation $g$. The last equality comes from the following computation of the first variation of the log-partition function:

$$\delta \log Z[f;g] = \frac{\delta Z[f;g]}{Z[f]} \tag{16}$$

$$= \frac{1}{Z[f]} \sum_{\mathbf{x}} \exp'\left(f(\mathbf{x})\right) g(\mathbf{x}) \tag{17}$$

$$= \sum_{\mathbf{x}} \frac{\exp\left(f(\mathbf{x})\right)}{Z[f]} g(\mathbf{x}) \tag{18}$$

$$= \mathbb{E}_{\mathbf{x}\sim q_f}\, g(\mathbf{x}). \tag{19}$$

Analogous to a Hessian, we can differentiate Equation 15 again along a second independent variation $h$ of $f$ yielding a symmetric bilinear functional which we will write as $\delta^2 L[f;g,h]$. Note that the first term in equation 2 is linear in $f$ and thus has no curvature, so we only have to consider the log partition function itself:

$$\delta^2 L[f;g,h] := \left.\frac{\partial^2 L[f + \epsilon g + \varepsilon h]}{\partial\epsilon\partial\varepsilon}\right|_{(\epsilon,\varepsilon)=0} \tag{20}$$

$$= -\delta^2 \log Z[f;g,h] = -\delta\left\{\delta \log Z[f;g]\right\}[f;h] \tag{21}$$

$$= -\delta\left\{\frac{1}{Z[f]} \sum_{\mathbf{x}} \exp\left(f(\mathbf{x})\right) g(\mathbf{x})\right\}[f;h] \tag{22}$$

$$= \frac{\delta Z[f;h]}{Z^2[f]} \sum_{\mathbf{x}} \exp\left(f(\mathbf{x})\right) g(\mathbf{x}) - \frac{1}{Z[f]} \sum_{\mathbf{x}} \exp'\left(f(\mathbf{x})\right) g(\mathbf{x})h(\mathbf{x}) \tag{23}$$

$$= \frac{\delta Z[f;h]}{Z[f]} \cdot \mathbb{E}_{\mathbf{x}\sim q_f} g(\mathbf{x}) - \frac{1}{Z[f]} \sum_{\mathbf{x}} \exp\left(f(\mathbf{x})\right) g(\mathbf{x})h(\mathbf{x}) \tag{24}$$

$$= \mathbb{E}_{\mathbf{x}\sim q_f} h(\mathbf{x}) \cdot \mathbb{E}_{\mathbf{x}\sim q_f} g(\mathbf{x}) - \mathbb{E}_{\mathbf{x}\sim q_f} h(\mathbf{x})g(\mathbf{x}) \tag{25}$$

$$= -\mathrm{Cov}_{\mathbf{x}\sim q_f}\left(g(\mathbf{x}), h(\mathbf{x})\right). \tag{26}$$

Note that this functional is negative semi-definite for all $f$, i.e. $\delta^2 L[f;h,h] \leq 0$, meaning that the log-likelihood is a concave functional of $f$.

Using these results, we can now compute the Taylor expansion of the increment in log-likelihood $L$ from a change $f \to f + \delta f$ up to second order in $\delta f$:

$$\Delta L_f[\delta f] = \delta L[f;\delta f] + \frac{1}{2}\delta^2 L[f;\delta f, \delta f] \tag{27}$$

$$= \mathbb{E}_{\mathbf{x}\sim p}\delta f(\mathbf{x}) - \mathbb{E}_{\mathbf{x}\sim q_f}\delta f(\mathbf{x}) - \frac{1}{2}\mathrm{Var}_{\mathbf{x}\sim q_f}\delta f(\mathbf{x}). \tag{28}$$

As an aside, defining the functional derivative, $\frac{\delta J[f]}{\delta f(\mathbf{x})}$, of a functional $J$ implicitly by:

$$\sum_{\mathbf{x}} \frac{\delta J[f]}{\delta f(\mathbf{x})} g(\mathbf{x}) = \delta J[f; g], \tag{29}$$

we can formally define, by analogy with the parametric case, the Fisher Information "Matrix" (FIM) at $f$ as the following bilinear functional of two independent variations $g$ and $h$:

$$F[f; g, h] := -\sum_{\mathbf{y}, \mathbf{z}} \left[ \mathbb{E}_{\mathbf{x} \sim q_f} \frac{\delta^2 \log q_f(\mathbf{x})}{\delta f(\mathbf{y}) \delta f(\mathbf{z})} \right] g(\mathbf{y}) h(\mathbf{z}) \tag{30}$$

$$= \sum_{\mathbf{y}, \mathbf{z}} \frac{\delta^2 \log Z[f]}{\delta f(\mathbf{y}) \delta f(\mathbf{z})} g(\mathbf{y}) h(\mathbf{z}) \tag{31}$$

$$= \delta^2 \log Z[f; g, h]. \tag{32}$$

The only difference to the second-order variation of 2 computed in equation 20 would be that the expectation is taken under the model distribution, $q_f$, instead of the data distribution $p$. However, because the only term in $\log q_f(\mathbf{x})$ that is non-linear in $f$ is the log-partition functional, which is not a function of $\mathbf{x}$, this expectation plays no role in the computation and we get the result that the FIM is the same as the negative Hessian of the log-likelihood for these models.

## A.1 Application to Piecewise Constant Functions

Considering a weak learner such as

$$\delta f(\mathbf{x}) = \sum_{j=1}^{J} w_j \mathbf{1}_{X_j}(\mathbf{x}), \tag{33}$$

where the subsets $X_j$ are disjoint and cover the entire input space, $\mathcal{X}$, we have that

$$\mathbb{E}_{\mathbf{x} \sim q} \delta f(\mathbf{x}) = \sum_{\mathbf{x} \in \mathcal{X}} q(\mathbf{x}) \sum_{j=1}^{J} w_j \mathbf{1}_{X_j}(\mathbf{x}) \tag{34}$$

$$= \sum_{j=1}^{J} w_j \sum_{\mathbf{x} \in X_j} q(\mathbf{x}) = \sum_{j=1}^{J} w_j Q(X_j). \tag{35}$$

Similarly, making use of the fact that $\mathbf{1}_{X_i}(\mathbf{x}) \mathbf{1}_{X_j}(\mathbf{x}) = \delta_{ij} \mathbf{1}_{X_i}(\mathbf{x})$, we can compute

$$\mathbb{E}_{\mathbf{x} \sim q} \delta f^2(\mathbf{x}) = \sum_{\mathbf{x} \in \mathcal{X}} q(\mathbf{x}) \left( \sum_{j=1}^{J} w_j \mathbf{1}_{X_j}(\mathbf{x}) \right)^2 = \sum_{j=1}^{J} w_j^2 Q(X_j). \tag{36}$$

In fact, we can extend this to any ordinary function of $\delta f$:

$$\mathbb{E}_{\mathbf{x} \sim q} \, g\left(\delta f(\mathbf{x})\right) = \sum_{\mathbf{x} \in \mathcal{X}} q(\mathbf{x}) \sum_{j=1}^{J} \mathbf{1}_{X_j}(\mathbf{x}) g\left(\delta f(\mathbf{x})\right) \tag{37}$$

$$= \sum_{j=1}^{J} \sum_{\mathbf{x} \in X_j} q(\mathbf{x}) g(w_j) \tag{38}$$

$$= \sum_{j=1}^{J} g(w_j) Q(X_j), \tag{39}$$

where we made use of the fact that the $\mathbf{1}_{X_j}$ constitute a partition of unity:

$$1 = \sum_{j=1}^{J} \mathbf{1}_{X_j}(\mathbf{x}). \tag{40}$$

Finally, we can compute the increase in likelihood from a step $f \to f + \alpha \cdot \delta f$ as

$$L[f + \alpha \cdot \delta f] - L[f] = \mathbb{E}_{\mathbf{x} \sim p}\left[\alpha \cdot \delta f(\mathbf{x})\right] - \log Z[f + \alpha \cdot \delta f] + \log Z[f] \tag{41}$$

$$= \alpha \mathbb{E}_{\mathbf{x} \sim p} \delta f(\mathbf{x}) - \log \mathbb{E}_{\mathbf{x} \sim q_f} \exp(\alpha \delta f(\mathbf{x})) \tag{42}$$

$$= \alpha \sum_{j=1}^{J} w_j P\left(X_j\right) - \log \sum_{j=1}^{J} Q_f\left(X_j\right) \exp\left(\alpha w_j\right), \tag{43}$$

where in equation 42 we made use of the equality:

$$\log Z[f + \alpha \cdot \delta f] - \log Z[f] = \log \frac{\sum_{\mathbf{x}} \exp(f(\mathbf{x}) + \alpha \delta f(\mathbf{x}))}{Z[f]} = \log \sum_{\mathbf{x}} q_f(\mathbf{x}) \exp(\alpha \delta f(\mathbf{x})), \tag{44}$$

and of the result in equation 39 in the final step.

This result can be used to conduct a line search over the step size using training data and to estimate an increase in likelihood at each round of boosting for the purpose of early stopping, using validation data.

## B  Training Density Estimation Trees

Density Estimation Trees (DET) [Ram and Gray, 2011] model the density function as a piecewise constant function,

$$q(\mathbf{x}) = \sum_{j=1}^{J} v_j \mathbf{1}_{X_j}(\mathbf{x}), \tag{45}$$

where $X_j$ are given by a partitioning of the input space $\mathcal{X}$ induced by a binary tree and the $v_j$ are the density values associated with each leaf that, for the time being, we will only require to be such that $q(\mathbf{x})$ sums to one.

Ram and Gray [2011] proposes fitting DET models to directly minimize a generative objective, the Integrated Squared Error (ISE) between the data generating distribution, $p(\mathbf{x})$ and the model:

$$\min_{q \in \mathcal{Q}} \sum_{\mathbf{x} \in \mathcal{X}} (p(\mathbf{x}) - q(\mathbf{x}))^2 . \tag{46}$$

Noting that $q$ is a function as in Equation 45 and that $\bigcup_{j=1}^{J} X_j = \mathcal{X}$, we can rewrite this as

$$\min_{v_1,\ldots,v_J,X_1,\ldots,X_J} \quad \sum_{\mathbf{x} \in \mathcal{X}} p^2(\mathbf{x}) + \sum_{j=1}^{J} \sum_{\mathbf{x} \in X_j} \left(v_j^2 - 2v_j p(\mathbf{x})\right)$$
$$\text{s.t.} \quad \sum_{j=1}^{J} \sum_{\mathbf{x} \in X_j} v_j = 1 . \tag{47}$$

Since the first term in the objective does not depend on the model this optimization problem can be further simplified as

$$\min_{v_1,\ldots,v_J,X_1,\ldots,X_J} \quad \sum_{j=1}^{J} \left(v_j^2 V(X_j) - 2v_j P(X_j)\right)$$
$$\text{s.t.} \quad \sum_{j=1}^{J} v_j V(X_j) = 1 , \tag{48}$$

where $V(X)$ denotes the volume of a subset $X$. Solving this quadratic program for the $v_j$ we obtain the following optimal leaf values and objective:

$$v_j^* = \frac{P(X_j)}{V(X_j)}, \qquad\qquad \text{ISE}^*\left(X_1,\ldots,X_J\right) = -\sum_{j=1}^{J} \frac{P^2(X_j)}{V_f(X_j)} . \tag{49}$$

One can therefore grow a tree by greedily choosing to split a parent leaf with support $X_P$ into two leaves with supports $X_L$ and $X_R$ so as to maximize the following criterion:

$$\max_{X_L, X_R \in \mathcal{P}(X_P)} \frac{P^2(X_L)}{V(X_L)} + \frac{P^2(X_R)}{V(X_R)} - \frac{P^2(X_P)}{V(X_P)} . \tag{50}$$

## B.1 Maximum Likelihood

To maximize the likelihood,

$$\max_q \mathbb{E}_{\mathbf{x} \sim p} \log q(\mathbf{x})\,, \tag{51}$$

rather than the ISE one can use the same approach. Here the optimization problem to solve is:

$$\max_{v_1,\dots,v_J,X_1,\dots,X_J} \quad \sum_{j=1}^{J} P(X_j) \log v_j$$
$$\text{s.t.} \quad \sum_{j=1}^{J} v_j V(X_j) = 1\,. \tag{52}$$

This is, again, easy to solve for $v_j$ since it is separable over $j$ after removing the constraint using Lagrange multipliers. The optimal leaf values and objective are in this case:

$$v_j^* = \frac{P(X_j)}{V(X_j)}, \qquad L^*(X_1,\dots,X_J) = \sum_{j=1}^{J} P(X_j) \log \frac{P(X_j)}{V_f(X_j)}\,. \tag{53}$$

The only change is therefore to the splitting criterion which should become:

$$\max_{X_L,X_R \in \mathcal{P}(X_P)} P(X_L) \log \frac{P(X_L)}{V(X_L)} + P(X_R) \log \frac{P(X_R)}{V(X_R)} - P(X_P) \log \frac{P(X_P)}{V(X_P)}\,. \tag{54}$$

# C  Greedy Tree Based Multiplicative Boosting

In multiplicative generative boosting an unnormalized current density model, $\tilde{q}_{t-1}(\mathbf{x})$, is updated at each boosting round by multiplication with a new factor $\delta q_t^{\alpha_t}(\mathbf{x})$:

$$\tilde{q}_t(\mathbf{x}) = \tilde{q}_{t-1}(\mathbf{x}) \cdot \delta q_t^{\alpha_t}(\mathbf{x})\,. \tag{55}$$

For our proposed NRGBoost, this factor is chosen in order to maximize a local quadratic approximation of the log likelihood around $q_{t-1}$ as a functional of the log density (see Section 3). The motivation behind the greedy approach of Tu [2007] or Grover and Ermon [2017] is to instead make the update factor $\delta q_t(\mathbf{x})$ proportional to the likelihood ratio $r_t(\mathbf{x}) := p(\mathbf{x})/q_{t-1}(\mathbf{x})$ directly, which under ideal conditions would mean that the method converges immediately when choosing a step size $\alpha_t = 1$. In more realistic setting, however, this method has been shown to converge under conditions on the performance of the individual $\delta q_t$ as discriminators between real and generated data [Tu, 2007, Grover and Ermon, 2017, Cranko and Nock, 2019].

While in principle this desired $r_t(\mathbf{x})$ could be derived from any binary classifier that is trained to predict a probability of a datapoint being generated (e.g., by training it to minimize a strictly proper loss) and Tu [2007] does not prescribe any particular choice, Grover and Ermon [2017] propose relying on the following variational bound of an $f$-divergence to derive an estimator for this ratio:

$$D_f(P \| Q_{t-1}) \geq \sup_{u \in \mathcal{U}_t} \left[ \mathbb{E}_{\mathbf{x} \sim p}\, u(\mathbf{x}) - \mathbb{E}_{\mathbf{x} \sim q_{t-1}} f^*(u(\mathbf{x})) \right]\,. \tag{56}$$

Here $f^*$ denotes the convex conjugate of $f$. This bound is tight, with the optimum being achieved for $u_t^*(\mathbf{x}) = f'(p(\mathbf{x})/q_{t-1}(\mathbf{x}))$, if $\mathcal{U}_t$ is capable of representing this function. $(f')^{-1}(u_t^*(\mathbf{x}))$ can thus be interpreted as an approximation of $r_t(\mathbf{x})$.

Adapting this method to use trees as weak learners can be accomplished by considering $\mathcal{U}_t$ in Equation 56 to be defined by tree functions $u = 1/J \sum_{j=1}^{J} w_j \mathbf{1}_{X_j}$ with leaf values $w_j$ and leaf supports $X_j$. At each boosting iteration a new tree, $u_t^*$ can thus be grown to greedily optimize the lower bound in the r.h.s. of Equation 56 and setting $\delta q_t(\mathbf{x}) = (f')^{-1}(u_t^*(\mathbf{x}))$ which is thus also a tree with the same leaf supports and leaf values given by $v_j := (f')^{-1}(w_j)$. This leads to the seaprable optimization problem:

$$\max_{w_1,\dots,w_J,X_1,\dots,X_J} \sum_{j}^{J} \left[ P(X_j) w_j - Q(X_j) f^*(w_j) \right]\,. \tag{57}$$

Table 4: Comparison of splitting criterion and leaf weights for the different versions of boosting.

| | Splitting Criterion | Leaf Values (Density) |
|---|---|---|
| DiscBGM (KL) | $P \log (P/Q)$ | $P/Q$ |
| DiscBGM ($\chi^2$) | $P^2/Q$ | $P/Q$ |
| NRGBoost | $P^2/Q$ | $\exp (P/Q - 1)$ |

Note that we drop the iteration indices from this point onward for brevity. Maximizing over $w_j$ with the $X_j$ fixed we have that $w_j^* = f' \left( P(X_j)/Q(X_j) \right)$ which yields the optimal value

$$J^*(X_1, \ldots, X_j) = \sum_j \left[ P(X_j) f' \left( \frac{P(X_j)}{Q(X_j)} \right) - Q(X_j)(f^* \circ f') \left( \frac{P(X_j)}{Q(X_j)} \right) \right] \qquad (58)$$

that in turn determines the splitting criterion as a function of the choice of $f$. Finally, the optimal density values for the leaves are given by

$$v_j^* = (f')^{-1} (w_j^*) = \frac{P(X_j)}{Q(X_j)} . \qquad (59)$$

It is interesting to note two particular choices of $f$-divergences. For the KL divergence, $f(t) = t \log t$ and $f'(t) = 1 + \log t = (f^*)^{-1} (t)$. This leads to

$$J_{KL}(X_1, \ldots, X_j) = \sum_j P(X_j) \log \frac{P(X_j)}{Q(X_j)} \qquad (60)$$

as the splitting criterion. The Pearson $\chi^2$ divergence, with $f(t) = (t-1)^2$, leads to the same splitting criterion as NRGBoost. Note however that for NRGBoost the leaf values for the multiplicative update of the density are given by $\exp \left( P(X_j)/Q(X_j) - 1 \right)$ instead of the ratio directly. Table 4 summarizes these results.

Another interesting observation is that a DET model can be interpreted as a single round of greedy multiplicative boosting starting from a uniform initial model. The choice of the ISE as the criterion to optimize the DET corresponds to the choice of Pearson's $\chi^2$ divergence and likelihood to the choice of KL divergence.

# D Implementation Details

**Discretization** In our practical implementation of tree based methods we first discretize the input space by binning continuous numerical variables by quantiles. Furthermore we also bin discrete numerical variables in order to keep their cardinalities smaller than 256. This can also be interpreted as establishing a priori a set of discrete values to consider when splitting on each numerical variable and is done for computational efficiency, being inspired by LightGBM [Ke et al., 2017].

**Categorical Splitting** For splitting on a categorical variable we once again take inspiration from LightGBM. Rather than relying on one-vs-all splits we found it better to first order the possible categorical values at a leaf according to a pre-defined sorting function and then choose the optimal many-vs-many split as if the variable was numerical. The function used to sort the values is the leaf value function. E.g., for splitting on a categorical variable $x_i$ we order each possible categorical value $k$ by $\hat{P}(x_i=k, X_{-i})/\hat{Q}(x_i=k, X_{-i})$ in the case of NRGBoost where $X_{-i}$ denotes the leaf support over the remaining variables.

**Tree Growth Strategy** We always grow trees in best first order. I.e., we always split the current leaf node that yields the maximum gain in the chosen objective value.

**Line Search** As mentioned in Section 3, we perform a line search to find the optimal step size after each round of boosting in order to maximize the likelihood gain in Equation 43. Because evaluating multiple possible step sizes, $\alpha_t$, is inexpensive, we simply do a grid search over 101 different step sizes in the range $[10^{-3}, 10]$ with their logarithm uniformly distributed.

Table 5: Dataset Information. We respect the original test sets of each dataset when provided, otherwise we set aside 20% of the original dataset as a test set. 20% of the remaining data is set aside as a validation set used for hyperparameter tuning.

| Abbr | Name | Train + Val | Test | Num | Cat | Target | Cardinality |
|------|------|-------------|------|-----|-----|--------|-------------|
| AB | Abalone | 3342 | 835 | 7 | 1 | Num | 29 |
| CH | California Housing | 16512 | 4128 | 8 | 0 | Num | Continuous |
| PR | Protein | 36584 | 9146 | 9 | 0 | Num | Continuous |
| AD | Adult | 32560 | 16280* | 6 | 8 | Cat | 2 |
| MBNE | MiniBooNE | 104051 | 26013 | 50 | 0 | Cat | 2 |
| MNIST | MNIST (downsampled) | 60000 | 10000* | 196 | 0 | Cat | 10 |
| CT | Covertype | 464810 | 116202 | 10 | 2 | Cat | 7 |

* Original test set was respected.

**Random Forest Density Estimation (RFDE)**   We implement the RFDE method [Wen and Hang, 2022] after quantile discretization of the dataset and therefore split at the midpoint of the discretized dimension instead of the original one. When a leaf support has odd cardinality over the splitting dimension a random choice is made over the two possible splitting values. Finally, the original paper does not mention how to split over categorical domains. We therefore choose to randomly split the possible categorical values for a leaf evenly as we found that this yielded slightly better results than a random one vs all split.

**Code**   Our implementation of the proposed tree-based methods is mostly Python code using the NumPy library [Harris et al., 2020]. We implement the tree evaluation and Gibbs sampling in C, making use of the PCG library [O'Neill, 2014] for random number generation.

# E   Datasets

We use 5 datasets from the UCI Machine Learning Repository [Dheeru and Karra Taniskidou, 2017]: Abalone, Physicochemical Properties of Protein Tertiary Structure (referred to as Protein in the sequence), Adult, MiniBooNE and Covertype. We also use the California Housing dataset which was downloaded through the Scikit-Learn package Pedregosa et al. [2011] and a downsampled version of the MNIST dataset Deng [2012]. Table 5 summarizes the main details of these datasets as well as the approximate number of samples used for train/validation/test for each cross-validation fold.

# F   Experimental Setup

## F.1   XGBoost Hyperparameter Tuning

To tune the hyperparameters of XGBoost we use 100 trials of random search with the search space defined in Table 6.

Table 6: XGBoost hyperparameter tuning search space. $\delta(0)$ denotes a point mass distribution at 0.

| Parameter | Distribution or Value |
|-----------|------------------------|
| `learning_rate` | LogUniform $\left(\left[10^{-3}, 1.0\right]\right)$ |
| `max_leaves` | Uniform $\left(\{16, 32, 64, 128, 256, 512, 1024\}\right)$ |
| `min_child_weight` | LogUniform $\left(\left[10^{-1}, 10^{3}\right]\right)$ |
| `reg_lambda` | $0.5 \cdot \delta(0) + 0.5 \cdot$ LogUniform $\left(\left[10^{-3}, 10\right]\right)$ |
| `reg_alpha` | $0.5 \cdot \delta(0) + 0.5 \cdot$ LogUniform $\left(\left[10^{-3}, 10\right]\right)$ |
| `max_leaves` | 0 (we already limit the number of leaves) |
| `grow_policy` | `lossguide` |
| `tree_method` | `hist` |

Each model was trained for 1000 boosting rounds on regression and binary classification tasks. For multi-class classification tasks a maximum number of 200 rounds of boosting was used due to the larger size of the datasets and because a separate tree is built at every round for each class. The best model was selected based on the validation set, together with the boosting round where the best performance was attained. The test metrics reported correspond to the performance of the selected model at that boosting round on the test set.

## F.2 TVAE Hyperparameter Tuning

To tune the hyperparameters of TVAE we use 50 trials of random search with the search spaces defined in Table 7.

The TVAE implementations used are from the latest version of the SDV package (`https://github.com/sdv-dev/SDV`) available at the time.

Table 7: TVAE hyperparameter tuning search space. We set both `compress_dims` and `decompress_dims` to have the number of layers specified by `num_layers`, with `hidden_dim` hidden units in each layer. We use larger batch sizes and smaller number of epochs for the larger datasets (MBNE, MNIST, CO).

| Parameter | Datasets | Distribution or Value |
|---|---|---|
| `epochs` | AB, CH, PR, AD | Uniform $([100..500])$ |
| | MBNE, MNIST, CO | Uniform $([50..200])$ |
| `batch_size` | AB, CH, PR, AD | Uniform $(\{100, 200, \ldots, 500\})$ |
| | MBNE, MNIST, CO | Uniform $(\{500, 1000, \ldots, 2500\})$ |
| `embedding_dim` | all | Uniform $(\{32, 64, 128, 256, 512\})$ |
| `hidden_dim` | all | Uniform $(\{32, 64, 128, 256, 512\})$ |
| `num_layers` | all | Uniform $(\{1, 2, 3\})$ |
| `compress_dims` | all | `(hidden_dim,) * num_layers` |
| `decompress_dims` | all | `(hidden_dim,) * num_layers` |

## F.3 TabDDPM Hyperparameter Tuning

To tune the hyperparameters of TabDDPM we use 50 trials of random search with the same search space that the original authors use in their paper [Kotelnikov et al., 2022].

We use the official implementation (`https://github.com/yandex-research/tab-ddpm`) adapted to use our datasets and validation setup.

## F.4 Random Forest Density Estimation

For RFDE models we train a total of 1000 trees. The only hyperparameter that we tune is the maximum number of leaves per tree for which we test the values $[2^6, 2^7, \ldots, 2^{1}4]$. For the Adult dataset, due to limitations of our tree evaluation implementation we only values test up to $2^{1}3$.

## F.5 Density Estimation Forests Hyperparameter Tuning

We train ensembles with 1000 DET models. Only three hyperparameters are tuned, using three nested loops. Every loop runs over the possible values of a single parameter in a pre-defined order with early stopping triggering if a value fails to improve the validation metric over the previous one. The tuned parameters along with their possible values are reported in Table 8

## F.6 NRGBoost

We train NRGBoost models for a maximum of 200 rounds of boosting. The starting point of each NRGBoost model was selected as a mixture model between a uniform distribution (10%) and the

Table 8: DEF models grid search space. Rows are in order of outermost loop to innermost loop. Note that for the Adult dataset, due to limitations of the implementation a maximum number of 8192 leaves is used instead of 16384.

| Parameter | Description | |
|---|---|---|
| max_leaves | The maximum number of leaves per tree | $[16384, 4096, 1024, 256]$ |
| feature_frac | The fraction of features to consider when splitting a node as a function of the total number of features $d$ | $[d^{-1/2}, d^{-1/4}, 1]$ |
| min_data_in_leaf | The minimum number of data points that need to be left in each leaf for a split to be considered | $[0, 1, 3, 10, 30]$ |

product of training marginals (90%) on the discretized input space. We observed that this mixture coefficient does not have much impact on the results however.

We only tune two parameters for NRGBoost Models:

- The maximum number of leaves for which we try the values $[64, 256, 1024, 4096]$ in order, stopping if performance fails to improve from one value to the next. For the CT dataset we also include 16384 in the values to test.
- The constant factor by which the optimal step size determined by the line search is shrunk at each round of boosting. This is essentially the "learning rate" parameter. To tune it we perform a Golden-section search for the log of its value using a total of 6 evaluations. The range we use is $[0.01, 0.5]$.

This means that at maximum we train only 24 NRGBoost models (30 for CT).

All other relevant parameters are fixed and their values, along with a short description, is given in Table 9.

Table 9: NRGBoost fixed parameters.

| Parameter | Description | |
|---|---|---|
| num_rounds | Total number of rounds of boosting | 200 |
| splitter | How the next leaf to split is determined | best |
| line_search | Whether to use a line search in determining the step size | True |
| max_ratio_leaf | Maximum ratio between training data and model data in each leaf | 2 |
| num_samples | Total number of samples in the sample pool | 80000 |
| | | 320000 (CT) |
| p_refresh | Indepdendent probability that a sample from the pool is replaced | 0.1 |
| burn_in | Number of samples to discard from the beginning of each chain | 100 |
| num_chains | Number of independent chains used for sampling | 16 |
| | | 64 (CT) |

## F.7   Evaluation Setup

**Single variable inference**   For the single variable inference evaluation, the best models are selected by their discriminative performance on a validation set. The entire setup is repeated five times with different cross-validation folds and with different seeds for all sources of randomness except on the CT dataset due to its large size. For the Adult and MNIST datasets the test set is fixed but training and validation splits are still rotated.

**Sampling**   For the sampling evaluation we use a single train/validation/test split of the real data (corresponding to the first fold in the previous setup) for training the generative models. The density models used are those previously selected based on their single variable inference performance on the validation set. For the sampling models (TVAE and TabDDPM) we directly evaluate their

ML Efficiency using the validation data by training an XGBoost model on generated data. The hyperparameters used for this XGBoost model are those selected on the real data in the previous experiment. We only use a generated validation set in order to select the best stopping point for XGBoost.

**ML Efficiency**    For each selected model we sample a train and validation sets with the same number of samples as those used on the original data. For NRGBoost we generate these samples by running 64 chains in parallel with 100 steps of burn in and downsampling their outputs by 30 (for the smaller datasets) or 10 (for MBNE, MNIST and CT). The setup is repeated 5 times with 5 different datasets generated for each method.

**Discriminator Measure**    We create the training, validation and test sets to train an XGBoost model to discriminate between real and generated data using the following process:

- The original validation set is used as the real part of the training set in order to avoid benefitting generative methods that overfit their training set.

- The original test set is split 20%/80%. The 20% portion is used as the real part of the validation set and the 80% portion as the real part of the test set.

- To form the generated part of the training, validation and test sets for the smaller datasets we sample data according to the original number of samples in the train, validation and test splits on the real data. Note that this makes the ratio of real to synthetic data 1:4 in the training set. This is deliberate because for these smaller datasets the original validation has few samples and adding extra synthetic data helps the discriminator.

- For the larger datasets we generate the same number of synthetic samples as there are real samples on each split, therefore making every ratio 1:1 because the discriminator is typically already too powerful and doesn't need extra data.

Because, in contrast to the previous metric, having a lower number of effective samples helps rather than hurts we take extra precautions to not generate correlated data with NRGBoost. We draw each sample by running its own independent chain for 100 steps starting from an independent sample from the initial model which is a rather slow process. The setup is repeated 5 times with 5 different sets of generated samples from each method.

## F.8   Computational Resources

The experiments were run on a machine equipped with an AMD Ryzen 7 7700X 8 core CPU and 32 GB of RAM. The comparisons with TVAE and TabDDPM further made use of a GeForce RTX 3060 GPU with 12 GB of VRAM.

# G   Additional Results

## G.1   Standard Errors

In Tables 10, 11 and 12 we report the sample standard deviations obtained for the main tables presented in the paper.

## G.2   Samples

In Figure G.2 we show the convergence of a Gibbs sampler sampling from a NRGBoost model. In only a few samples each chain appears to have converged to the data manifold after starting at a random sample from the initial model (a mixture between the product of training marginals and a uniform). Note how consecutive samples are autocorrelated. In particular it can be rare for a chain to switch between two different modes of the distribution (e.g., switching digits) even though a few such transitions can be observed.

Table 10: Single variable inference sample standard deviations.

| | $R^2$ | | | AUC | | Accuracy |
|---|---|---|---|---|---|---|
| | AB | CH | PR | AD | MBNE | MNIST |
| XGBoost | 0.0354 | 0.0092 | 0.0036 | 0.0004 | 0.0005 | 0.0017 |
| RFDE | 0.0963 | 0.0039 | 0.0071 | 0.0023 | 0.0078 | 0.0101 |
| DEF (ISE) | 0.0373 | 0.0080 | 0.0023 | 0.0026 | 0.0108 | 0.0107 |
| DEF (KL) | 0.0271 | 0.0083 | 0.0038 | 0.0005 | 0.0009 | 0.0073 |
| NRGBoost | 0.0358 | 0.0113 | 0.0087 | 0.0006 | 0.0007 | 0.0009 |

Table 11: ML Efficiency results sample standard deviations.

| | $R^2$ | | | AUC | | Accuracy | |
|---|---|---|---|---|---|---|---|
| | AB | CH | PR | AD | MBNE | MNIST | CT |
| TVAE | 0.0059 | 0.0054 | 0.0054 | 0.0011 | 0.0002 | 0.0088 | 0.0013 |
| TabDDPM | 0.0182 | 0.0049 | 0.0072 | 0.0007 | 0.0000 | 0.0250 | 0.0012 |
| DEF (KL) | 0.0131 | 0.0063 | 0.0073 | 0.0011 | 0.0022 | 0.0283 | 0.0029 |
| NRGBoost | 0.0161 | 0.0010 | 0.0076 | 0.0009 | 0.0009 | 0.0008 | 0.0011 |

Table 12: Discriminator measure sample standard deviations.

| | AB | CH | PR | AD | MBNE | MNIST | CT |
|---|---|---|---|---|---|---|---|
| TVAE | 0.0039 | 0.0055 | 0.0017 | 0.0012 | 0.0001 | 0.0000 | 0.0001 |
| TabDDPM | 0.0146 | 0.0045 | 0.0043 | 0.0022 | 0.0024 | 0.0000 | 0.0074 |
| DEF (KL) | 0.0129 | 0.0081 | 0.0022 | 0.0016 | 0.0000 | 0.0000 | 0.0001 |
| NRGBoost | 0.0167 | 0.0115 | 0.0059 | 0.0032 | 0.0005 | 0.0026 | 0.0058 |

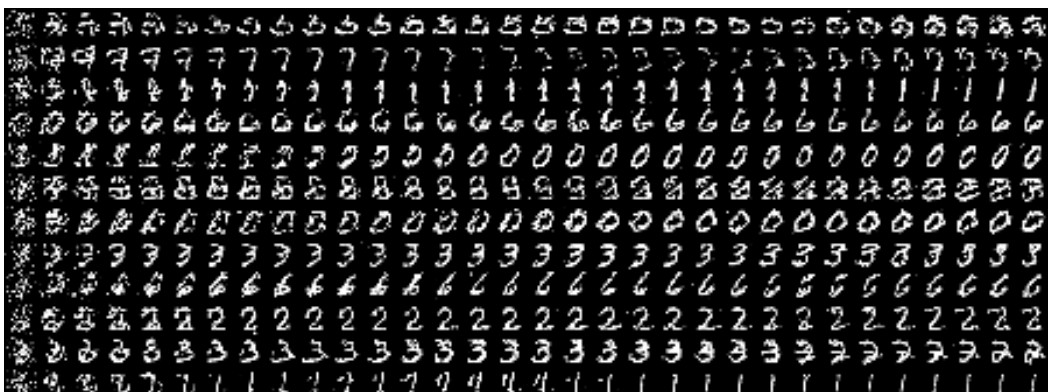

Figure 3: Downsampled MNIST samples generated by Gibbs sampling from a NRGBoost model. Each row corresponds to an independent chain initialized with a sample from the initial model $f_0$ (first column). Each column represents a consecutive sample from the chain.

Table 13: Best NRGBoost model parameters per dataset and the wall time taken to train it. The format is minutes:seconds.

|  | AB | CH | PR | AD | MBNE | MNIST | CT |
|---|---|---|---|---|---|---|---|
| max_leaves | 64 | 1024 | 1024 | 256 | 1024 | 4096 | 16384 |
| shrinkage | 0.14 | 0.063 | 0.14 | 0.09 | 0.199 | 0.199 | 0.098 |
| Time | 1:18 | 4:17 | 5:27 | 3:54 | 20:36 | 149:30 | 179:11 |

## G.3 Time

In Table 13 we report the best hyperparameters found for NRGBoost for the first cross-validation fold together with the time taken to train this best model.

