# OpenReview forum: "NRGBoost: Energy-Based Generative Boosted Trees"
_NeurIPS.cc/2024/Conference — Submitted to NeurIPS 2024_

### Official Review · Reviewer_fQFZ · 2024-06-20

**Soundness:** 4
**Presentation:** 4
**Contribution:** 4
**Rating:** 9
**Confidence:** 4

**Summary:**

The authors propose a energy-based generative boosting method. They try to maximize the log-likelihood functional delta with a second-order expansion. This lead to a boosting algorithm with deltas as steps in the log-likelihood. Instead of scaling the steps with a fixed predefined value (as a anti-overfitting hyperparameter), they rely on linear search to get better performance. They initialize f0 as uniform and use trees as weak learner to learn the deltas. They derive the objective with trees, it is well explained and looks similar to other second-order method objectives like XGBoost. They use MCMC to sample from Q(x) as is typical from energy-based approaches. They use Gibbs so they only need to sample from one dimension while keeping the rest constant. However since there are t trees, this is quadratic in t. They use some form of accept/reject to sometimes accept previous iteration samples in order to not have to re-samples new samples given the quadratic cost. They include a probability of refresh in order to not just always accept old samples. They propose interesting ways of regularizing the approach. They provide good literature review of related methods. It seems like Section 4 could be integrated with the related work section.

Figure may appear unimpressive to a generative expert unfamiliar with trees. But, being able to generate good samples from MNIST using only decision is an impressive feat (even if MNIST is downsampled). This paves the way for trees being used on more complex data.

Table 2 shows nice prediction results, and they do extensive hyperparameter tuning. However, ML Efficiency is not the best metric, it only focus on prediction. A generative model could produce low-diversity fake data that lead to good classifiers. I would recommending adding some distribution metrics such as the Wasserstein distance (which is computable in low-dimension and is used for high-dim data such as images/videos in the form of the widely popular FID/FVD). There are also other useful metrics for quality and diversity, I recommend looking the extensive choice of tabular-data metrics described in https://proceedings.mlr.press/v238/jolicoeur-martineau24a.html. However, the only essential one in my opinion is having a distance in distribution. It can be the Wasserstein distance or something like the MMD distance.

Except for the missing distribution metric, everything else in the results is good and the methodology is sound with good hyperparameter tuning. I would encourage the authors to add a distribution metrics.

The method is sound, novel and quite interesting. The presentation is very well made. Being the first tree method achieving good image data samples is impressive in my opinion. This is a very high quality paper.

**Strengths:**

The method is sound, novel and quite interesting.

The presentation is very well made.

Being the first tree method achieving good image data samples is impressive in my opinion.

This is a very high quality paper.

**Weaknesses:**

Missing a distribution metric, the ML efficiency is not a adequate metric on its own.

**Questions:**

What is the memory requirements, especially when processing large datasets such as MNIST? Could you provide sample time and RAM requirements for the datasets? How hard is scaling with respect to N_samples and also scaling with respect to N_features?

**Limitations:**

If the authors had to downsample MNIST from 28x28 to lower, than there are scaling limitations that should be added.

---

> ### Author Rebuttal · Authors · 2024-08-05
>
> We thank the reviewer for their valuable feedback and encouraging words about our work, as well as for providing helpful context about our method's ability to produce passable image data as a tree-based generative model.
> We address the reviewer's two concerns below.
>
> # Distribution Metric
>
> While we did not include a conventional distribution metric, our original intention was to have the discriminator measure reported in the paper serve a similar purpose.
> Our reasoning was that estimation of integral probability metrics (like the Wasserstein-1 distance or kernel MMD) between two distributions can be interpreted as solving a binary classification problem to distinguish between the two sets of samples (see e.g., [[Sriperumbudur et al., 2012]](https://projecteuclid.org/journals/electronic-journal-of-statistics/volume-6/issue-none/On-the-empirical-estimation-of-integral-probability-metrics/10.1214/12-EJS722.full)).
> Therefore, instead of picking a classifier belonging to a specific family of functions (such as continuous functions with Lipschitz constant <= 1 for the $W_1$ metric), we chose instead to directly employ a classifier that we know to be state of the art for binary classification in tabular data and that has no issue with the data being mixed categorical and numerical with very different scales.
>
> That being said, we agree that this evaluation is perhaps unconventional and our approach has downsides. Namely, that XGBoost is too effective and can nearly perfectly distinguish real from synthetic data on the larger datasets where it has enough training data.
> We therefore decided to follow the reviewer's suggestion and evaluate the Wasserstein distance between samples using a similar setup to the linked paper:
> - Numerical variables are min-max scaled and categoricals are one-hot encoded and scaled by 1/2.
> - A $L_1$ distance is used in the optimal transport formulation.
> - Due to the worst case cubic scaling with the sample size of solving the optimal transport problem we sub-sample a maximum of 5000 samples from the original test set and use an equal number of synthetic samples.
>
> The full results for the test set are added to the PDF attached to the author rebuttal but below we summarize the average rank over all datasets and 5 experiments per dataset for convenience:
> |method|$\hat{W}_{train}$|$\hat{W}_{test}$|
> |-|-|-|
> |TVAE|3.114|3.143|
> |TabDDPM|2.743|2.771|
> |ARF|2.743|2.771|
> |DEF (KL)|4.090|4.057|
> |NRGBoost|2.314|2.257|
>
> We note however an almost complete inversion of the ranking of the methods compared to our discrimination measure results in the abalone and miniboone datasets.
> Another somewhat concerning observation is the instability of the results with respect to different, seemingly equally plausible choices of how to normalize or rescale the data and to the distance metric used in the OT formulation.
>
> Despite this, NRGBoost still ranks best on average even though these results don't favor it as much as the discriminator measure.
> A possible explanation for the comparatively worse results is that NRGBoost (and DEF KL) are trained to minimize the KL-divergence between data and model distributions which behaves quite differently from a Wasserstein distance.
> Regardless, we believe that both types of evaluation are valuable and provide different perspectives of the results and trade-offs between the different methods.
> We will add the full results, together with the above discussion to the paper as we agree it would improve it  significantly. We kindly ask the reviewer to please let us know if they have any further feedback regarding this evaluation.
>
> # Scalability Concerns
>
> Memory was never a concern during our experiments with any of our tree-based methods. Besides the memory required to store the original data itself and the similarly sized sample pool, **NRGBoost uses no significant additional memory** besides storing the trees themselves which use a very compact format.
>
> The choice to downsample MNIST was simply one of computational time.
> Both the tree-fitting and the sampling scale linearly with the number of features. Downscaling by 4x therefore allowed us to run the experiments with cross-validation on MNIST in a reasonable time-frame.
> Note that training a NRGBoost model with 200 trees of 4096 leaves used to take roughly 2.5 hours and we trained 6 of those (+ smaller ones) per cross-validation fold (i.e. x5).
> We have since made significant improvements to the efficiency of our tree-fitting implementation which was previously responsible for the bulk of the training time and are now able to train the same model on downsampled MNIST in under 1 hour.
> Gibbs sampling now represents roughly 70% of training time which we report for each dataset in the PDF attached to the author's rebuttal.
> This **sampling time scales linearly with both `N_samples` and `N_features`.**
>
> We note however that downsampling also helps with the curse of dimensionality when it comes to density estimation.
> Tabular models like NRGBoost do not benefit from any of the helpful inductive biases of convolutional neural networks and treats every pixel as a feature that is potentially correlated to any other feature without any awareness of spatial separation between pixels.
> Increasing the resolution by 4x means that the density over 4x as many features needs to be learned and to achieve similar visual accuracy we expect that this would require deeper trees as there are more features to split on and partition the input space.
> Because our primary goal was to model tabular data we did not explore directions that would improve NRGBoost on image data specifically by exploiting the biases mentioned above but that could be interesting future work!
>
> # Summary
> Again, we thank the reviewer for their time and their suggestions and hope to have adequately addressed their concerns.

---

> > ### Comment · Reviewer_fQFZ · 2024-08-07
> > **response**
> >
> > I agree that the Discriminator score can be linked to a IPM metric loosely, but having a true metric like Wasserstein is better. I agree that there is no perfect way to normalize/scale the data when you have continuous and categorical data, this is a well known problem. The Gower distance specifically tries to tackle this problem and this the distance that you used for the Wasserstein distance (by min-max continuous and /2 one-hot), which at least makes the choice rational.
> >
> > Thank you for addressing my requests for a distribution metric and memory concerns. Given this, I am increasing my score by 1.

---

> ### Author Response · Authors · 2024-08-07
> **Thank you!**
>
> Thank you once again for your time in reviewing our rebuttal and for increasing your score.
>
> We are glad we were able to address your concerns.

---

### Official Review · Reviewer_UJVb · 2024-07-07

**Soundness:** 2
**Presentation:** 2
**Contribution:** 2
**Rating:** 4
**Confidence:** 2

**Summary:**

This paper proposes a boosted tree algorithm that performs distribution learning using an energy-based formulation. Inspired by methods like XGBoost, it is claimed to achieve high performance not only in generative ability but also in discriminative performance.

**Strengths:**

The proposed method incorporates techniques that can be leveraged because they inherit from tree boosting models, such as the approximate sampling algorithm. Moreover, as a model that possesses not only generative quality but also discriminative performance, it has a wide range of applications.

**Weaknesses:**

This study is not theoretical; therefore, the validity of the proposed method must be confirmed through experiments. However, there are some questions regarding the experimental settings.

Also, from an algorithmic perspective, since methods unique to tree ensembles are incorporated, they could potentially offer advantages in terms of computational complexity compared to other methods. However, evaluations regarding efficiency have not been conducted. If there are advantages, it seems opportunity loss.

Please refer to the Questions section.

Minor:
The evaluation of variance is presented separately in Appendix G, but it is difficult to compare. Therefore, I would like it to be summarized in one table.

**Questions:**

**(1) Robustness**

In the discussion, it is stated that hyperparameters are not critical compared to deep learning, but it seems that there is no experimental evidence to support this. If you claim robustness to hyperparameters as a strength, please provide evidence for this.

**(2) Computational cost**

Please describe the computational cost and compare it with benchmarks. In the discussion, DEF is mentioned to not require sampling for training and that the processing for sampling is light, but please quantify how much of a difference there is.

**(3) Dataset**

How do you select datasets? It seems that only a few are picked from among the many datasets available, which may give the impression that you are cherry-picking the data that works well.

**(4) Fixed parameters**

It seems that the hyperparameters are fixed in advance as shown in Table 9, but how were these decided? It is mentioned that the hyperparameter search can be covered by only 24 combinations, but this also gives the impression that you are setting them in advance to work well and narrowing the apparent search space.

**(5) Parameter range**

It was also mentioned that DEF tends to have a large number of leaves, reaching 2^14, but isn't this because the search range is set that way? Even if you align this setting with NRGBoost, which has a maximum of 2^12, will it lead to the same conclusion?

**(6) Benchmark performance**

In Figure 1, the performance of the benchmark methods seems too poor. Is this figure created with the same settings as those measured in Table 2, for example? TabDDPM seems to show better performance than the proposed method in the ML Efficiency result, but it is hard to imagine getting similar results from the generated results in Figure 1.

**Limitations:**

Computational cost of sampling is larger than existing methods.

---

> ### Author Rebuttal · Authors · 2024-08-05
>
> We thank the reviewer for their detailed feedback. We will try to address all their concerns below.
> Regarding the minor point about separately reporting averages and the respective standard deviations, we agree that this was a bad choice and will change the paper to present them in the same format used in the PDF attached to the author's rebuttal.
> # (1) Robustness
> We concede to the reviewer's point that we did not provide evidence for the claim of robustness as it was based on our own analysis and interpretation of the results of the hyperparameter tuning.
> We can add an appendix with this analysis which show that, for example, models with 256 or 1024 leaves and a shrinkage factor around 0.15 can both achieve reasonable performance across most datasets for NRGBoost.
> # (2) Computational Cost
> We have added a comparison of the time taken to fit the best models found by hyperparameter tuning for each dataset to the PDF attached to the author rebuttal above.
>
> Regarding the difference between sampling from NRGBoost and DEF models we are afraid that any numbers we can present for DEF would be misleading.
> Since sampling is required during training for NRGBoost, we have spent considerable effort re-writing it in C and optimizing it.
> In contrast, for DEF, sampling is a one-off cost that we incur for a single model after hyperparameter tuning and as a result it is implemented in single threaded Python code.
> When we claim that sampling is simpler for DEF models it is because it requires only picking a tree in the ensemble uniformly at random, then a leaf according to a stored vector of leaf probabilities, and finally sampling a point uniformly from the support of the leaf. None of these steps require any complex calculations compared to the Gibbs sampling and an optimized implementation should be **many** times faster.
> # (3) Dataset
> Running our cross-validation setup for all density methods with hyperparameter tuning for each method/fold takes considerable time.
> As a result, we had to limit the number of datasets used and we believe that 7 datasets is a reasonable number and in line with other papers on the subject of generative modelling (e.g., [[Xu et al., 2019]](https://arxiv.org/abs/1907.00503) and
> [[Watson et al., 2023]](https://arxiv.org/abs/2205.09435)).
>
> We tried to strike a good balance between tasks, #samples and #features. Furthermore, almost all of our datasets have appeared previously in other works on synthetic sample generation such as those mentioned above and [[Kotelnikov et al., 2022]](https://arxiv.org/abs/2209.15421). We can provide more specific reasoning behind the choice of each particular dataset in case the reviewer is interested.
> # (4) Fixed Hyperparameters
> For training NRGBoost we tune only the maximum number of leaves per tree and the shrinkage factor since they have the largest impact on the overall results.
> The remaining parameters reported in Table 9 are indeed kept fixed because we wanted to have a simpler and less time consuming hyperparameter tuning.
>
> However, note that some of these entries are not really parameters but rather algorithmic choices inspired by regular boosting.
> The remaining are parameters that we found to work well enough, be somewhat redundant or otherwise not have a large impact in results.
> This was mostly determined over the course of developing the method, either in early experiments on toy data or one-off runs in a single dataset/fold.
> While we acknowledge that we do not report this extra experimentation, it is because of its limited nature and the fact that once the method was crystallized we never really tried to experiment with changing these parameters.
> Tuning some of these (like the maximum ratio in each leaf or increasing the number of rounds of boosting) may actually improve results at the cost of a more expensive tuning.
> Due to space limitations, we fully explain the rationale behind the choice of each of these parameters in a separate comment in case the reviewer is interested.
> # (5) Parameter Range
> We believe that the description of our tuning setup in section F.5 may have been misleading and we will make it clearer in the paper.
>
> In fact, we always train DEF models with $2^{12}$ leaves (varying all the inner loop parameters). We only stop hyperparameter tuning if none of these models is better than the best model with $2^{14}$ leaves (the early stopping works per parameter and only exits the respective inner parameter loop if that makes sense). As a result, we know that decreasing the number of leaves did not yield a better DEF model.
> Please let us know if this answers your question or if you would like us to further clarify this point.
> # (6) Benchmark Performance
> The samples shown in Figure 1 for TabDDPM were generated from the same model and were part of the samples used to compute the ML efficiency results in Table 2 (**note that TabDDPM is second to last on the MNIST dataset**).
> The problem is that we have simply not been able to achieve reasonable results with TabDDPM on MNIST even after trying different normalization strategies, significantly increasing the number of training steps or the number of diffusion steps.
> Most pixel features on this dataset are bimodal with most samples being either 0 or 255 and very few falling in between. The numerical data generated by TabDDPM is consistently outside the expected range, and is squashed into the two extremes of the range when inverting the input normalization leading to random looking images.
> That being said, we share your concerns that the results as presented in the paper may be misleading and we plan to add a disclosure of our inability to obtain a reasonable model with TabDDPM for this dataset.
> # Summary
> We would like to thank the reviewer for their time and we hope to have adequately addressed all of their concerns. We would kindly ask them to let us know if anything wasn't clear and also to check the other improvements made to the paper listed in the author rebuttal above.

---

> > ### Comment · Reviewer_UJVb · 2024-08-09
> >
> > Thank you for your response. I would like to discuss complexity.
> >
> > I believe that differences due to variations in implementation (e.g., programming languages) are not important. If all benchmarks are not implemented consistently during evaluation, the comparison becomes unequal, and no claim of superiority can be made. Additionally, it seems that implicit parallelization is also being done through techniques like sampling, so it is difficult to objectively understand the actual complexity by only looking at the computation time.
> >
> > Since your paper is empirical, I believe a fair assessment can only be made by properly evaluating the trade-off between performance and cost.
> >
> > Could you please describe the complexity of the proposed method and the benchmarks using Big-O notation and compare them? This might better demonstrate the validity of the proposed method rather than just computational speed.
> > According to the Rebuttal PDF, there are significant variations across different datasets. What is the cause of this?

---

> ### Author Response · Authors · 2024-08-05
> **Fixed Hyperparameters**
>
> Below we explain our rationale for the choice of the fixed hyperparameters for NRGBoost:
> - `num_rounds`: In earlier versions of the method, before we introduced importance sampling and rewrote our Gibbs sampling implementation in C, going above 200 rounds was prohibitevely expensive. While increasing this will tend to always improve results as long as the shrinkage parameter is also being tuned (and using early stopping), we decided to keep 200 rounds as the overall cost of sampling during training still scales quadratically with this number. Furthermore, this value is in line with values that already work well for discriminative boosting algorithms like XGBoost or LightGBM (which has a default of 100).
> - `splitter`: Inspired by LightGBM's default approach to always split the leaf that leads to the best improvement in the objective. This allows more expressive models with fewer leaves than alternative approaches like depthwise splitting (XGBoost default) which wastes splits on leafs that don't really lead to much improvement.
> - `line_search`: As explained in the paper this typically allows the model to converge faster by taking larger steps in the beginning of training where the quadratic approximation we optimize is farther from the likelihood. Incidently we find that the step size tends to settle closer to 1 as training progresses which is the optimal step size predicted by the quadratic approximation we try to optimize.
> - `max_ratio_leaf`: We found early on with toy data that limiting this to small values tends to work better for regularization. Since this also limits the growth of the energy function at each round and therefore reduces the number of samples rejected from the pool we found no reason to increase it. In the end, this plays a similar role to the shrinkage parameter which we already tune so it was a bit redundant to tune both.
>
> As for the sampling parameters, we have always used a similar number of samples to the original training set and this has always worked as a reasonable rule of thumb. Going over that rarely improved the results by much. But since it also never hurts to have more samples and the smaller datasets trained fast enough already, we kept the number 80000 fixed as a rough upper bound to the training set size for all datasets except for covertype (for which we used 4 times more to roughly match its #samples in training).
>
> We chose to use 16 MCMC chains since these run in parallel and that is the number of virtual cores in the CPU used to run the experiments. For covertype we also multiplied the number of chains by 4 to keep the #samples/chain similar.
>
> For the `burn_in` and `p_refresh` parameters we just picked seemingly reasonable values as they don't seem to have a large impact (other than `p_refresh` increasing the sampling time which is why we chose to set it at 10%).
>
> We hope that this clarifies the choices made. Please let us know if you have any further questions.

---

> ### Author Response · Authors · 2024-08-07
> **Additional comments on MNIST results**
>
> We thought we should provide some additional comments on the performance of benchmark methods on MNIST which we could not fit in the rebuttal due to space limitations.
>
> NRGBoost and the other methods we compare to are meant for **tabular data**. As such, they do not benefit from the helpful inductive biases of convolutional neural networks and have no awareness of the spatial relation between pixels.
> They are in fact invariant to a permutation of the pixels and need to learn the complex relationships between every pixel and all other pixels.
> The dimensionality of this dataset is also higher than other typical benchmarks for tabular data.
> All these properties make this problem very challenging for tabular models and (as reviewer **fQFZ** also seems to agree) we consider it a big achievement for NRGBoost to do as well as it does.
> We do not believe that other methods underperforming on this dataset is unexpected as it's not really the type of data that tabular models are designed for. Still, we thought it would be interesting to include it as it allows us to visually assess actual samples from the model.
>
> Still, regarding the performance of competing methods, we note that downsampled MNIST was also used as a benchmark in [[Xu et al., 2019]](https://arxiv.org/abs/1907.00503) which introduces TVAE and CTGAN. However, samples from neither of the methods are ever shown in the paper.
> On the request of reviewer **trn6**, we have also included a comparison to a tree-based generative model (ARF) which seems to perform better than our original baselines (please refer to the PDF attached to the author's rebuttal).
>
> Finally, ML Efficiency can be very deceptive about the ability of the model to accurately capture the input distribution, $p(x)$.
> The most important property to achieve a high score in ML efficiency is that the model captures well the relationship between the target $y$ variable and the remaining input variables $x$ (i.e., has a good internal model for $p(y \vert x)$ since this is what the downstream discriminative model is trying to estimate).
> Modeling well the remaining $p(x)$ (which is what is responsible for the images themselves) is secondary because a wrong $p(x)$ only means that the samples are inefficiently placed for optimal learning of the discriminative function.
> Reviewer **fQFZ** makes a similar point in their review that models can produce low diversity data that still lead to good classifiers.
>
> We note also that it is surprisingly easy to score well in **classification** on MNIST. Input features (i.e. pixels) are very redundant for predicting $y$. Even just modeling the relationship between a small subset of pixels and the target variable y can be enough to achieve a much better than random accuracy. As an example, using only the **8 pixels** in a central vertical line XGBoost is already able to achieve an accuracy of 0.649. Therefore a model presumably doesn't even need to model well the relationship between $y$ and all pixels, just a small subset in order to achieve a good ML efficiency score.
>
> We hope this clarifies some of the reviewer's doubts about the results on this dataset.

---

> ### Author Response · Authors · 2024-08-09
> **Response to Reviewer Comment (1/2)**
>
> We thank the reviewer for their time in reviewing our rebuttal and for being willing to discuss our paper.
> Below we outline the computational complexities of all methods but note that these provide only asymptotic guidance.
>
> ## Tree-based methods
> All tree based methods have to fit the ensemble of trees. The time complexity of this should be similar across all non-random methods (ARF, DEF, NRGBoost). Essentially, it scales as $O(NFTD)$ where $N$ is the number of data points, $F$ the number of features, $T$ the number of trees in the ensemble and $D$ the depth of the trees with the following caveats:
> - For NRGBoost the factor $N$ should be replaced by $N + M$ where $M$ is the total number of samples in a pool of samples (see below). As mentioned previously in our comment on fixed hyperparameters, choosing $M$ similar $N$ works well in practice so the overall scaling **with the data characteristics** is still $O(NF)$.
> - For ARF there is an additional multiplicative factor which is the number of times the adversarial loop runs. This is a complex function of the dataset as the algorithm stops when no improvement can be made. In practice we observe that it is somewhere between 2 and 6 iterations with smaller datasets generally stopping earlier.
> ### Sampling in NRGBoost
> To fit a tree at each round of boosting, NRGBoost requires samples from the model. We keep a pool of samples of size $M$ and before each round of boosting we need to resample a portion of those which we will call $m$ and assume that it is roughly constant across rounds. Note that this sampling has nothing to do with implicit parallelization like the reviewers' statement seems to imply. It is merely an additional required step of our algorithm which we happen to be able to parallelize effectively.
>
> The total cost of sampling is on average $O(mFT^2 D)$. As can be seen this process also scales linearly with $F$ and $N$ (through $m$). The quadratic scaling with the $T$ hyperparameter is the reason we only train NRGBoost models with 200 trees vs the DEF models that are trained with 1000 trees.
>
> **To summarize, the computational complexity of all tree-based methods scales linearly with both number of data points and number of features as far as the characteristics of the data are concerned.**
>
> As for the variation across different datasets, this is mostly due to these characteristics, both directly and indirectly:
> - **Directly:** Datasets with a larger $NF$ factor will be slower which is the main reason that MNIST is the slowest followed by either miniboone or covertype.
> - **Indirectly:** The characteristics of the dataset also change what are the best hyperparameters. For tree-based models, the number of leaves (depth $D$) is perhaps the biggest factor where datasets that favor larger trees will have slower training times for the best model, but other hyperparameters also have an effect.
>
> ## Neural Networks
> Computational complexity for neural network methods depends on the architecture of the network and type of method. While we are not experts on the matter we can make a few educated guesses.
>
> As an example, just the matrix multiplication in an input layer for $T$ training steps with a batch size $B$ would be $O(TBFH)$ where $H$ is the number of hidden output units of this layer.
> This is similar to tree-based methods' scaling with $NF$ when we assume that a fixed number of epochs are used to train the model (i.e., $T$ is proportional to $N/B$).
> Depending on the method there would be other multiplicative factors such as the number of diffusion timesteps and there are also the other layers of course.
> These other layers may be responsible for most of the training time in practice and their input and output sizes are hyperparameters that may be chosen not to scale with $F$.
> These models can also leverage massive parallelism from the use of GPUs which makes these comparisons based on computational complexity not very helpful in our opinion.
>
>
> # Summary
>
> We believe all methods scale similarly with the characteristics of the data under reasonable assumptions.
> However this does not tell the full picture as they also scale differently with the very different hyperparameters of each method.
> Controlling for computational time is difficult because there is hyperparameter tuning in the loop which might favor hyperparameters that lead to slower or faster models depending on the dataset.
>
> We will follow up with an additional comment justifying our benchmarking approach and our statements due to space limitations.

---

> ### Author Response · Authors · 2024-08-09
> **Response to Reviewer Comment (2/2)**
>
> We agree with the reviewer that a fair benchmarking of all methods is important for any claim of superiority.
> In regards to tree-based models, we believe that we gave every possible advantage to our original baselines:
> - We proposed DEF models in section 4 of our paper as an improvement over the original Density Estimation Trees (DET), introducing two algorithmic improvements (bagging and changing from ISE metric to KL divergence) in order to get them to perform well and to have a worthy baseline.
> - Overall, DEF models took **longer** to run hyperparameter tuning and single models are usually slower to train. Despite this they still achieve lower performance consistently.
> - NRGBoost tends to scale much better with depth of the trees than DEF models and in many datasets we could reduce the number of leaves with only a small performance penalty.
> In contrast, we had to push DEF models to the limit of what our implementation would allow to even have reasonable performance. If we trained DEF models with trees of only 1024 leaves (typical for NRGBoost) performance would be abysmal in general.
>
> Generally, the performance gap between NRGBoost and DEF models is so large that even if we were to further handicap NRGBoost by, e.g., reducing the number of trees and the maximum number of leaves it would still come out on top.
> As a result we believe that our original statement that NRGBoost performs better than the additive tree-based ensemble models is accurate.
>
>
> For neural network methods, because we are not experts, we chose to rely on guidance from the existing literature when it comes to hyperparameter tuning.
> But since these are such different methods, running on different types of hardware, and with very different hyperparameters and hyperparameter tuning requirements it can be particularly challenging to achieve an experimental setup that everyone will agree is fair. We try to find a setup that we believe is representative of what a practitioner may get when using these methods and, like for our tree-based baselines, generally try to err on the side of caution. As an example, we give neural networks methods the benefit of using ML Efficiency directly as the hyperparameter tuning metric which is one of the metrics they are later evaluated on.
>
> Nonetheless we are commited to being as fair as possible when presenting our results and we value the feedback from the reviewer on this issue. We welcome any further suggestions or recommendations on this matter that could improve our paper.
>
> Best regards,
>
> The authors

---

> > ### Comment · Reviewer_UJVb · 2024-08-12
> >
> > Thanks for your response. I have read your response.

---

> > > ### Author Response · Authors · 2024-08-12
> > >
> > > Thank you for aknowledging our response. We want to reiterate that we are open to discussing any remaining concerns you might have.

---

### Official Review · Reviewer_QQ7t · 2024-07-09

**Soundness:** 3
**Presentation:** 3
**Contribution:** 2
**Rating:** 6
**Confidence:** 4

**Summary:**

This paper proposed an energy based generative boosting algorithm analogous to XGBoost, which can be used as generative model as well as be applied to discriminative tasks.

**Strengths:**

The energy-based boosting is novel. The proposed method is capable of both generative sampling and discriminative tasks, enabling broad methodological applications.

**Weaknesses:**

My concerns regarding the proposed method and the experiments are detailed in the questions section.

**Questions:**

1. For Table 1:
Why is NRGBoost shown in bold when it's overperformed by XGBoost? In other words why is XGBoost used as a baseline?
Is there a particular reason for SE reported separately in the supplement?

2. For the sampling experiments in section 6.2, is the authenticity of synthetic samples measured in some ways? (In an extreme case, if the synthetic samples are almost replicating training data, they will achieve high ML efficiency and good discriminator measures)

4. How is the scalability?

5. Can the proposed method be used on (unordered) categorical data, which is very common in tabular data?

**Limitations:**

I do not identify significant limitations other than the ones discussed in the paper.

---

> ### Author Rebuttal · Authors · 2024-08-05
>
> We thank the reviewer for their feedback. Below are our responses to each individual question.
>
> # Question 1
>
> Our main goal is to compare generative methods and we mark as bold the best **generative** model on each dataset. We will add a note to the caption of Table 1 to make this clear since XGBoost was only meant to provide a reference for the best discriminative performance that could reasonably be achieved by a good **discriminative** model on these datasets.
> In practice we expect generative models to have worse performance because discriminative models are trained to estimate $p(y \vert \mathbf{x})$ directly and not the more general joint distribution $p(y, \mathbf{x})$ which is a harder problem but also one that yields more flexibility in applications.
> We therefore believe that NRGBoost coming close to the XGBoost baseline is many datasets is already a positive result.
>
> Regarding the SEs being reported separately, we agree that this was a poor decision on our part. We will update all the tables in the main paper to report SE alongside the average values. Please check the PDF attached to the author's rebuttal for examples of the new formatting. Thank you for raising this issue.
>
> # Question 2
>
> In the context of generative models, memorizing the training data is an extreme case of overfitting where the model learns the empirical training distribution instead of approximating the idealized distribution that generates the data.
>
> Regarding the ML efficiency metric reported in Section 6.2, we share the reviewer's concerns that it is susceptible to favoring overfitted models since in the extreme case where the training data is memorized, a method would simply be able to achieve the same performance as that obtained by training XGBoost directly in the original training data.
> We note however that it is one of the most commonly used metrics when evaluating synthetic data (e.g.,
> [[Xu et al., 2019]](https://arxiv.org/abs/1907.00503),
> [[Kotelnikov et al., 2022]](https://arxiv.org/abs/2209.15421)).
>
> This is why we believe is important to have other metrics that are not as susceptible to overfitting and our discriminator measure setup was designed with this in mind.
> We train the discriminator (XGBoost) to distinguish between synthetic samples from the model and samples from held out data (validating also using held out data).
> Since the probability that the same sample appears both in the training set for the generative model and the held out data should be very small (or effectively zero if the distribution is at least in part continuous), this discriminator can simply learn to predict 1 for all synthetic data in its training set.
> This might seem like an overfitted discriminator but if the generative model was simply outputting a subset of the training samples, such a discriminator would be expected to also perform well in our held out test comprising of synthetic data and real held out data and therefore yield a poor discriminator measure.
>
> Finally, we note also that for density models, overfitting to the empirical training distribution would also likely cause poor single variable inference results in the same way an overfitted discriminative model would.
> As a result, at least for density models, we can check overfitting by comparing train and test single variable inference results and we can also check for methods that over-perform in ML Efficiency when compared to their single variable inference results.
>
> # Question 3
>
> The computational and memory cost of fitting the trees in NRGBoost should, in theory, be the same as for any other tree-based model such as XGBoost, with the caveat that we have a larger dataset comprising both real data and a pool of samples from the model. The scaling of this tree-fitting with the number of samples is linear as well as with the number of features, rounds of boosting and depth of the trees.
>
> The main computational cost of our algorithm, however, is the Gibbs sampling that needs to be performed before every round of boosting to resample a fraction of the samples in our sample pool.
> As mentioned in the paper, at round $t$ of boosting this cost scales in the worst case as $O(tNFm)$ with $m$ being the number of samples, $F$ being the number of features and $N$ being the number of internal nodes in each tree (in practice only a small subset of nodes needs to be transversed for each sample).
> The total cost of sampling over $T$ rounds of boosting therefore scales as $O(T^2NFm)$, assuming that $m$ samples are drawn per round.
> The memory cost of sampling is not a real concern as besides the space required to store the samples (and the trees) we make no significant use of extra memory.
>
> In the author's rebuttal we add the training times required for each of the models selected by hyperparameter tuning on each dataset which can be used to gauge how this scaling plays out in practice as larger datasets and datasets with more features typically also benefit from larger models with deeper trees.
>
> # Question 4
>
> Naturally handling categorical data is one of the advantages of tree-based models and both NRGBoost and DEF models are both able to do it without requiring any tricks such as one-hot encoding or other forms of encoding.
> In all of our experiments we used many-vs-many categorical splits at the tree nodes similar to how LightGBM splits categorical data.
> We have also implemented one-vs-all splits in our code but have found that these tend to perform slightly worse when using trees of similar depth. 4 of the 7 datasets we use in the experiments section have (multi-class) categorical variables.
>
> # Summary
>
> We hope to have answered all of the reviewer's questions about our paper in a satisfactory manner and are happy to provide additional clarifications if necessary.
> We would also like to kindly ask the reviewer to consider the other improvements made to the paper that were enumerated in the author's rebuttal above and thank the reviewer for their time.

---

> ### Author Response · Authors · 2024-08-10
> **Additional Experiment on Sample Memorization**
>
> We apologize for the additional comment but the reviewer's question about memorizing the training data gave us an idea about an additional experiment that we believe supports our argumentation above and might interest the reviewer.
>
> We introduce a **memorizing** baseline that simply memorizes a fraction of the training set.
> When prompted for synthetic samples, it samples from this memorized set with replacement.
> We try two settings for this baseline:
> - Memorizing 10% of the training set
> - Memorizing the full training set
>
> Finally we evaluate the same discriminator setup we use in the paper.
> This experiment allows us to gauge how well learning the training empirical distribution (with different sample sizes) performs.
>
> Below we report the average discriminator measure results we obtained when repeating each experiment 5 times as in the paper.
> We also include the results for the two best methods (TabDDPM and NRGBoost) for ease of comparison.
> Note that these results are ROC AUC for distinguishing synthetic from real (held out) samples and as such **lower is better**.
>
>
> | Dataset         | AB | CH | PR |  AD  | MBNE | MNIST    | CT |
> |-----------------|---------|------------|---------|---------|-----------|---------|-----------|
> | TabDDPM         |   0.818 |      0.667 |**0.628**|  0.604  |     0.789 |  1.000  |     0.915 |
> | NRGBoost        |**0.625**|   **0.574**|   0.631 |**0.559**|     0.993 |  0.943  |     0.724 |
> | Memorize (10%)  |   0.996 |      1.000 |   0.995 |  1.000  |     0.959 |  0.960  |     0.958 |
> | Memorize (100%) |   0.763 |      0.799 |   0.762 |  0.769  |  **0.607**|**0.611**|  **0.606**|
>
> For the smaller datasets, even the best possible memorization baseline fails to beat the best method on each respective dataset.
> For the larger datasets (MiniBooNE, MNIST and Covertype), while memorizing the full training set does beat all other methods, memorizing a more sensible fraction of it is not enough.
> As we increase the memorized set, the empirical distribution becomes closer to the data generating distribution and therefore discriminator measure results improve but we hope that this shows that simply memorizing samples is often not enough to achieve best performance on this metric due to the failure to generalize.
>
> We would like to kindly ask the reviewer if their concerns have been resolved and if not, what remaining concerns they have.
> We would greatly appreciate this feedback in order to further improve our paper.
>
> Best regards,
>
> The authors

---

> > ### Comment · Reviewer_QQ7t · 2024-08-10
> >
> > Thanks for the rebuttal and additional experiments. This addressed my concerns and I increase my score.

---

> > > ### Author Response · Authors · 2024-08-11
> > > **Thank you**
> > >
> > > We are glad we were able to address your concerns. Thank you for raising your score.

---

### Official Review · Reviewer_trn6 · 2024-07-12

**Soundness:** 3
**Presentation:** 2
**Contribution:** 3
**Rating:** 5
**Confidence:** 2

**Summary:**

The paper proposes to extend the success of tree-based methods in discriminative tasks to generative modelling, which is implemented via an energy-based generative boosting algorithm (NRGBoost). Specifically, NRGBoost directly extends the tree-based tabular models by replacing the discriminative objectives with a generative one, which seems novel to me.

**Strengths:**

1. The paper has well-founded rationales: (1) Tree-based models are performant in discriminative tasks, and thus they are highly likely to also be performant in generative tasks on tabular data, and (2) existing tree-based generative methods do not preserve the tree structures well.
2. The paper is well-written, especially the notations.

**Weaknesses:**

1. **[Important]** Some highly relevant benchmark methods are missing, including ARF (tree-based) [1], GOGGLE (diffusion) [2] and TabPFGen (energy-based) [3].

2. In Line 325, the authors claim that the proposed method “significantly” outperforms other methods, while the significance test seems missing.

3. I would suggest the authors add comparison results on the computation efficiency. Because NRGBoost basically employs the same architecture as traditional gradient boosting trees, the computation efficiency should be higher than most other network-based generative models.

4. There seem to be some typos throughout the main text: “I.e.” (Line 272)

5. **[Important]** Code is not provided. I remain conservative about the results claimed in the paper.

[1] Watson, David S., et al. "Adversarial random forests for density estimation and generative modeling." International Conference on Artificial Intelligence and Statistics. PMLR, 2023.

[2] Liu, Tennison, et al. "GOGGLE: Generative modelling for tabular data by learning relational structure." The Eleventh International Conference on Learning Representations. 2023.

[3] Ma, Junwei, et al. "TabPFGen–Tabular Data Generation with TabPFN." NeurIPS 2023 Second Table Representation Learning Workshop.

**Questions:**

1. How does NRGBoost perform single variable inference (Lines 271-277)?

**Limitations:**

The authors detail the limitations of NRGBoost in Section 7.

---

> ### Author Rebuttal · Authors · 2024-08-05
>
> Thank you for the thoughtful and careful review of our work and for the detailed feedback provided.
> We will try to address all the weaknesses pointed out below.
> # Missing Methods
> **ARF:**
> As another tree-based density method we agree that this is a very valuable comparison to make. We have thus made the effort to implement single variance inference for this method ourselves since the official python version of ARF did not implement density evaluation.
> We have included the main results with this method in the PDF attached to the author's rebuttal above.
>
> **GOGGLE:**
> We were not aware of GOGGLE or TabPFGen and we thank the reviewer for bringing these methods to our attention.
> We agree that GOGGLE would make an interesting comparison. We briefly tried to use the implementation available through the `synthcity` python library for the convenience of having a library interface but quickly ran into issues with package versions even after installing the library in a fresh virtual environment.
> Given the short time-frame of the rebuttal period, the uncertainty of what other issues we might encounter and our unfamiliarity with the method we decided it was best to focus our efforts on other areas of improvement rather than rush to have results ready in only a few days. So while we can't have these results ready for the rebuttal we will do our best to do this comparison as well, provided we do not run into any further serious issues.
>
> **TabPFGen:**
> As for TabPFGen, we did not find an available implementation which makes it difficult to obtain a comparison in such a short time frame. However, after taking a closer look at the paper, we also believe that this method, while interesting, would not be a good fit for our current evaluation setup for a couple of reasons:
>   - It requires a categorical target which makes it incompatible with our regression scenarios.
>   - The authors of [[Ma et al., 2024]](https://arxiv.org/abs/2406.05216) acknowledge that dataset size is a limitation due to the TabPFN's inference step's inability to deal with large datasets as input. The largest datasets used in [Ma et al., 2024] have 2000 instances which leads us to believe that this method would be impractical on the datasets that we use as it would require significant downsampling.
>   - Finally, as explained in our experiments section, our setup hinges on the training of the generator being agnostic to what the target variable is. For all the generative models that we train, $y$ is not conditioned on and is treated as any other input variable. But in TabPFGen, the $y$ variable is generated directly from the $p(y|x)$ of a pre-trained classifier (TabPFN) which would make it unfair when evaluating single variable inference or ML efficiency over the same $y$ variable compared to the other methods. We also did not compare to [[Correia et al., 2020]](https://arxiv.org/abs/2006.14937) due to the same concern.
> Note also that, similarly to TabPFGen, when a preferred $y$ variable exists, a generative model can always be used to learn only $p(x)$ while a state of the art discriminative model can be used to provide the $p(y|x)$ (e.g., XGBoost). This is another reason why we chose to focus on general fitness of the $p(y, x)$ instead.
> # Significance Test for Single Variable Inference
> While our original intention with the statement was merely to comment on the large gap in the results we agree that it is easy to misinterpret.
> We have therefore verified that a paired t-test rejects the null hypothesis of equal means when comparing NRGBoost to all the other density methods (including now ARF) in Table 1 on all datasets (at a confidence level of 95%).
> Of additional note is that the same test fails to reject this null hypothesis when comparing NRGBoost to XGBoost for the abalone (p=0.329), california (p=0.679) and protein (p=0.709) datasets.
> We can add additional tables to appendix G with the p-values if the reviewer believes it adds value.
> # Computational Efficiency
> We have added a comparison of the time taken to fit the best models found by hyperparameter tuning for each dataset to the PDF attached to the author's rebuttal above.
> Note that while the tree fitting part of the algorithm should, in principle, be comparable to other tree-based discriminative algorithms such as XGBoost, NRGBoost requires additional Gibbs sampling at the beginning of each boosting round.
> After recent improvements to our tree fitting code, sampling now represents the biggest cost of training an NRGBoost model (~70% on average by our estimates). Note however that:
> - There is still margin for optimization of our tree-fitting code which is not nearly as optimized as a mature implementation such as XGBoost.
> - Gibbs sampling can extract more benefit from parallelization if a higher core count CPU is used.
> # Typos
> We thank the reviewers for raising this point. We spell-checked the text before submission but have also detected a few typos and other small issues since then. We will thoroughly check the text again.
> # Code
> We have sent an anonymized link with our code to the AC so that they may share with reviewers as per the instructions we have received for code sharing.
> # Single Variable Inference
> Given an energy function $f(y,\mathbf{x})$ for $q_f(y,\mathbf{x})$, $q_f(y\vert\mathbf{x})$ can be computed as:
> $$q_f(y\vert\mathbf{x})=\frac{\exp f(y,\mathbf{x})}{\sum_{y^\prime}\exp f(y^\prime,\mathbf{x})}$$
> This is essentially a softmax, involving only computing $\exp f(y, \mathbf{x})$ for all possible values of $y$ (which can be done efficiently for a tree based model) and normalizing.
> This could definitely be clearer in the paper but we had to cut this explanation to meet the page limits. We will try to fit it back in in a future version of the paper if given the opportunity.
>
> We would like to thank the reviewer for their time and we hope to have adequately addressed all of their concerns. We are happy to provide additional clarifications if necessary.

---

> ### Author Response · Authors · 2024-08-12
> **GOGGLE comparison**
>
> We apologize for the extra comment but given the fast approaching end of the discussion period we thought it would be better to antecipate any questions the reviewer might have and provide an update on our ongoing efforts to add GOGGLE as a baseline as the reviewer originally requested.
>
> We have managed to get a working version of the synthcity library with the GOGGLE plugin (and we can't stress enough that this was not an easy task).
> So far, we have only managed to train models with (mostly) the library's default hyperparameters on 5 of the datasets but, in the process, have faced significant challenges which we outline below.
>
> # Slow training
> The method is considerably slower than all the other methods we compare:
> - As an extreme example, training a model for 500 epochs on `covertype` took **more than 12 hours** at the maximum batch size used in the paper (128).
> Note that the paper (and the library default) uses 1000 epochs (with early stopping) which could lead to more than 1 full day to train a single model!
> - We haven't been able to run GOGGLE on the `MiniBooNE` or `MNIST` datasets as it seems that it **scales poorly with the number of features in the dataset**.
> For context, **a single epoch on the `MiniBooNE` dataset takes 17 minutes to complete**.
> - Even for the datasets where we are able to train models, the slow training times render us unable to do hyperparameter tuning in a reasonable timeframe like we do for all the other methods.
>
> # Lackluster Performance
> While we would like to explore the hyperparameter space for this model better, the results we have obtained so far with the default hyperparameters have yielded poor performance in general.
>
> On all of our regression datasets we consistently get $R^2$ values below 0 for the ML efficiency metric.
> While we are suspicious of these results and will need to investigate further we note that the paper appears to focus on binary classification tasks so we have no frame of reference.
>
> On classification datasets, trained models tend to only generate data from majority classes:
> - On the adult dataset, using our training setup, the trained model exclusively produces samples of the majority class.
> - On the covertype dataset, it also rarely produces samples of any of the classes outside of the top two (out of 7 classes). This makes performance on this dataset poor as well.
> - Even if we could conditionally generate based on the class, this goes against the idea of our setup which, as we explained above, is meant to produce models that are good for any inference task over any potential variable, not just a specific one. It would, furthermore, make it unfair to all other methods for which we don't use conditional generation (even though we could for some of them, including NRGBoost).
>
> # Conclusion
> We will continue to investigate our options and explore different choices of hyperparameters to try to achieve a setup that we believe is fair, both to GOGGLE and the other methods.
> As it stands, given the long training times, we do not believe we will be able to provide a meaningful comparison with this baseline; at least not in any of the larger datasets.
> However we would very much welcome any suggestions or recommendations from the reviewer on this subject.
>
> We would also be very thankful if the reviewer could provide any feedback on whether their concerns have been adequately addressed and offer again to clarify any remaining questions that they might have.
> We have made a significant effort to comply with every request of the reviewer to the best of what we believe is reasonably possible and we think that this has significantly improved our paper as a result. In light of this we would kindly ask the reviewer to consider re-evaluating their score.
>
> Best regards,
>
> The authors

---

> > ### Comment · Reviewer_trn6 · 2024-08-13
> > **Response to rebuttal**
> >
> > Thank you for your detailed response and the efforts you've put into addressing the concerns raised. After carefully considering your points and other reviewers' comments, I still believe that my initial evaluation remains valid: a fair and complete benchmark should be necessary before acceptance, so my score remains unchanged.

---

> ### Author Response · Authors · 2024-08-13
>
> We thank the reviewer for their time and regret that our addition of the ARF comparison was not sufficient to address their concerns about a complete benchmark.
>
> We decided to focus our efforts on ARF because it is also a density estimator and thus, the more interesting baseline given that it could be used in the same types of applications as NRGBoost. Unfortunately, this required implementing single variable inference for this method ourselves which consumed a significant portion of our rebuttal week (together with running hyperparameter tuning on 5 cross-validation folds for every dataset).
>
> For the reasons we explained above, we don't believe that TabPFGen would be an appropriate comparison for our setup, even if we were able to run it on datasets of the size that we use.
>
> Furthermore, assuming that we could resolve the performance issues we are currently experiencing with GOGGLE, due to the method being so slow to train we would have to severely limit the number of epochs relative to what was used in the original paper in order to bring training times in line with other methods. We will try to add a comparison along these lines in the future but we are not sure how valuable it will be given these constraints.
>
> Best regards,
>
> The authors

---

### Author Rebuttal · Authors · 2024-08-06

We thank all reviewers for taking the time to review our work.
We have been working diligently to incorporate their feedback in order to improve the paper.
Below is a list of the main changes we have done as a result.
We kindly ask the reviewers to also check the attached PDF which includes additional results reflecting these changes.
# Comparison to ARF
We added a comparison to the tree-based density method **ARF** [[Watson et al., 2023]](https://arxiv.org/abs/2205.09435) as requested by reviewer **trn6**.
We implemented our own density evaluation for this method in order to be able to compare to the remaining density methods on our single variable inference task.
The hyperparameter tuning performed is similar to DEF models where the maximum number of leaves per tree and minimum number of samples per node are tuned using grid search.

Because ARF's single variable inference results were close to NRGBoost's on the covertype dataset we also made the effort to run 5-fold cross-validation on this dataset (for all density methods).
We believe that the overall results obtained (reported in the attached PDF) help solidify NRGBoost's claim as the top performing generative model capable of density estimation.
# Wasserstein Distance
We added an additional metric for comparing synthetic samples (Wasserstein distance) as recommended by reviewer **fQFZ**.
We follow the same setup used in [[Jolicoeur-Martineau et al., 2024]](https://proceedings.mlr.press/v238/jolicoeur-martineau24a.html) for its evaluation.

While the results (reported in the attached PDF) don't favor NRGBoost as heavily as our reported discriminator measure, it still ranks the best on average across all generated datasets.
Please refer to the rebuttal to reviewer **fQFZ** for a more in-depth discussion about this metric.
# Standard Deviation Reporting
As recommended by reviewers **QQ7t** and **UJVb** we will change all tables in the paper to report standard deviations alongside mean values (in the same format as the attached PDF).
# Computational Efficiency Improvements and Comparison
We have made significant improvements to our tree-fitting code, greatly reducing the training times originally reported for NRGBoost in Table 13.
Fitting the trees was previously the most time consuming part of training a NRGBoost model but now it represents only ~30% of the training time on average across all the datasets.
While we believe there is still plenty of margin for improvement, we are now much more confident that the new training times are representative of what an optimal implementation can achieve.

We report these new training times for NRGBoost as well as for every other method for the best model selected by hyperparameter tuning in the attached PDF.
A few caveats about these results though:
- We do not report the training times for the RFDE method because our implementation is sub-optimal and it should, in theory, be virtually free when compared to the other methods. This is because the splitting process is random and depends only on the input domain. The data itself is only required for computing leaf probabilities which is inexpensive.
- Many of the optimizations done for tree-fitting of NRGBoost models unfortunately do not extend to our implementation of DEF models. As a result, we don't believe the numbers we report for this method are very representative of what an optimal implementation could do.
- Currently, the biggest computational cost for training a NRGBoost model is the Gibbs sampling. This process can efficiently leverage higher parallelism than what we used in the experiments (16 virtual cores).
# Fix erroneous results for TVAE method
We realized that due to a configuration change while debugging, only the first 10 trained models were considered during model selection for the TVAE method in the MNIST and covertype datasets.
We have fixed this issue resulting in better performance for this method in the ML Efficiency metric but no (visible) change in our discriminator measure to the precision that is reported.

We hope that these changes resolve most of the remaining concerns about our work.


Many thanks,

The authors

---

### Decision · Program_Chairs · 2024-09-25

**Decision:**

Reject

**Comment:**

The authors propose tree-based methods for generative modeling.
- **Reviews**: The reviews highlight several weaknesses, particularly criticizing the choice of datasets, fixed hyperparameters, and the absence of comparative methods (in particular, well-known generative alternatives for tabular data are missing).
- **Rebuttal**: The authors provided new results and included one of the suggested methods by reviewer `trn6` for comparison.
- **Reviewer-AC Discussion Phase**: During this phase, reviewers `UJVb` and `trn6` expressed concerns. While their scores are borderline (reject and accept, respectively), they indicate that the authors' rebuttal did not address all open issues and that their most significant concerns remain unresolved. Furthermore, both explicitly state that they believe the paper, in its current form, should not be accepted.
- **Decision**: While the paper is well-written, addresses an interesting problem, and has an above-average score, the latter is primarily due to the score by Reviewer `fQFZ` and, as `UJVb` noted, might not reflect the consensus among reviewers. Given that two out of four reviewers recommend against accepting the paper in its current form, I recommend rejecting the paper.